# Specific interaction of an RNA-binding protein with the 3′-UTR of its target mRNA is critical to oomycete sexual reproduction

Hui Feng[1,2,3], Chuanxu Wan[1,2,3], Zhichao Zhang[1,2,3], Han Chen[1,2,3], Zhipeng Li[1,2,3], Haibin Jiang[1,2,3], Maozhu Yin[1,2,3], Suomeng Dong[1,2,3], Daolong Dou[1,2,3], Yuanchao Wang[1,2,3], Xiaobo Zheng[1,2,3], Wenwu Ye[1,2,3]*

1 Department of Plant Pathology, Nanjing Agricultural University, Nanjing, Jiangsu, China, 2 The Key Laboratory of Plant Immunity, Nanjing Agricultural University, Nanjing, Jiangsu, China, 3 The Key Laboratory of Integrated Management of Crop Diseases and Pests (Ministry of Education), Nanjing, Jiangsu, China

* yeww@njau.edu.cn

## Abstract

Sexual reproduction is an essential stage of the oomycete life cycle. However, the functions of critical regulators in this biological process remain unclear due to a lack of genome editing technologies and functional genomic studies in oomycetes. The notorious oomycete pathogen *Pythium ultimum* is responsible for a variety of diseases in a broad range of plant species. In this study, we revealed the mechanism through which PuM90, a stage-specific Puf family RNA-binding protein, regulates oospore formation in *P. ultimum*. We developed the first CRISPR/Cas9 system-mediated gene knockout and *in situ* complementation methods for *Pythium*. *PuM90*-knockout mutants were significantly defective in oospore formation, with empty oogonia or oospores larger in size with thinner oospore walls compared with the wild type. A tripartite recognition motif (TRM) in the Puf domain of PuM90 could specifically bind to a UGUACAUA motif in the mRNA 3′ untranslated region (UTR) of *PuFLP*, which encodes a flavodoxin-like protein, and thereby repress *PuFLP* mRNA level to facilitate oospore formation. Phenotypes similar to *PuM90*-knockout mutants were observed with overexpression of *PuFLP*, mutation of key amino acids in the TRM of PuM90, or mutation of the 3′-UTR binding site in *PuFLP*. The results demonstrated that a specific interaction of the RNA-binding protein PuM90 with the 3′-UTR of *PuFLP* mRNA at the post-transcriptional regulation level is critical for the sexual reproduction of *P. ultimum*.

## Author summary

Oomycetes are a class of eukaryotic microorganisms with life cycles and growth habits similar to filamentous fungi, but are not true fungi. Although sexual reproduction, which produce oospores, is an essential stage of life cycle, the functions of critical regulators in this biological process remain unclear due to a lack of genome editing technologies and functional genomic studies in oomycetes. In this study, we developed the first CRISPR/ Cas9 system-mediated gene knockout and *in situ* complementation methods for *Pythium*

**Data Availability Statement:** All relevant data are within the manuscript and its Supporting information files. The RNA-seq data have been deposited in the National Center for Biotechnology

Information (NCBI) database [BioProject ID: PRJNA540115 (Run IDs: SRR15049129-SRR15049146)].

**Funding:** This work was supported by grants to W. Y. from the National Natural Science Foundation of China (NSFC; 31972250 and 31772140), grants to Y. W. from the NSFC (31721004) and the China Agriculture Research System (CARS-004-PS14), and grants to H. F. from the Postgraduate Research & Practice Innovation Program of Jiangsu Province (KYCX20_0594). The funders had no role in study design, data collection and analysis, decision to publish, or preparation of the manuscript.

**Competing interests:** The authors have declared that no competing interests exist.

*ultimum*, a notorious oomycete pathogen that is responsible for a variety of diseases in a broad range of plant species. We further identified the Puf family RNA-binding protein PuM90 and the flavodoxin-like protein PuFLP as major functional factors involved in *P. ultimum* oospore formation. We proposed a new model that PuM90 acts as a stage-specific post-transcriptional regulator by specifically binding to the 3′-UTR of *PuFLP* and then repressing *PuFLP* mRNA level. This study describes new technologies and data that will help to elucidate sexual reproduction and post-transcriptional regulation in oomycetes.

## Introduction

Oomycetes are a class of eukaryotic microorganisms with life cycles and growth habits similar to filamentous fungi, but are not true fungi [1]. Unlike fungi, oomycetes produce oospores during sexual reproduction [2–4]. In both homothallic and heterothallic oomycete species, fertilization results in thick-walled zygotes called oospores, which can be produced within infected plant tissue and released into the soil as the plant tissue degrades. Oospores are highly resistant to environmental influences and can persist in the soil for several years [4]. The sexual cycle enhances fitness through formation of recombinant genotypes that may be more pathogenic or resistant to fungicides than their parents [5]. Consequently, understanding the biological processes and molecular mechanisms that regulate sexual reproduction is essential for precise and efficient prevention of plant disease outbreaks associated with these pathogens.

One of the most destructive oomycete pathogens [6], *Pythium ultimum*, displays a wide host range and has been recorded on hundreds of plant species worldwide, causing damping-off and root rot that lead to huge yield losses [7]. *P. ultimum* is homothallic, meaning that a single isolate can mate with itself to complete the sexual stage; however, outcrossing has also been reported [8]. *P. ultimum* includes two varieties: *P. ultimum* var. *ultimum* (the most common and pathogenic group) and *P. ultimum* var. *sporangiferum* (a rare and less pathogenic group); the former scarcely produces zoospores and almost solely produces oospores [9]. Therefore, sexual reproduction is likely more important in the life cycle of *P. ultimum* var. *ultimum* (hereinafter referred as *P. ultimum*).

To date, only a few genes involved in the sexual reproduction of oomycetes have been identified. In *Pythium oligandrum*, small tyrosine rich (PoStr) proteins play a key role in oospore formation. Oospores of *PoStr* family-silenced mutants displayed major ultrastructural changes and were sensitive to degradative enzyme treatment [10]. In *Phytophthora sojae*, knockout mutants of *PsYPK1*, a serine/threonine protein kinase-encoding gene, lost the ability to produce oospores [11]; and silencing mutants of *PsGK5*, one of the genes coding G-protein-coupled receptors with a phosphatidylinositol phosphate kinase domain, was also defective in oospore formation [12]. In *Phytophthora infestans*, silencing of a loricrin-like protein-encoding gene significantly affected oospore formation [13]. Despite the identification of the regulators above, the regulatory mechanisms of oospore formation and sexual reproduction process in oomycetes remain largely unknown.

The release of genome sequences has opened the door for elucidation of *P. ultimum* biology at the molecular level [14]. However, barriers for functional genomics remain, and one of the greatest is genetic manipulation technologies. Although protoplast electroporation and *Agrobacterium tumefaciens*-mediated transformation were proved to be available in *P. ultimum* [15,16], few studies have reported using these methods. In recent years, the CRISPR/Cas system has been applied to oomycetes such as *Phytophthora sojae* [17], *Phytophthora palmivora*

[18], *Phytophthora capsici* [19], *Phytophthora infestans* [20], *Aphanomyces invadans* [21], and *Peronophythora litchii* [22], greatly promoting functional genomics research. Although poly-ethylene glycol (PEG)-mediated protoplast transformation has been reported in a few *Pythium* species [23], CRISPR/Cas system-mediated genetic manipulation methods, such as gene knockout and *in situ* complementation, have not yet been reported in any *Pythium* species.

Regulation of gene expression at the post-transcriptional level, including RNA stability, translation, and localization, is important for the growth and development of eukaryotic organisms [24]. RNA-binding proteins are a major group of regulators that directly or indi-rectly participate in these processes [25]. Among them, Puf proteins, named for *Drosophila* Pumilio (PUM) and *Caenorhabditis elegans* fem-3 binding factor (FBF), share a Puf RNA-binding domain and are present in various eukaryotic species. Puf proteins are involved in diverse biological processes, including embryonic development [26], nervous system function and development [27], stem cell and germ cell maintenance [28], rRNA processing and ribo-some biogenesis [29], and chemotactic cell movement [30]. They commonly repress level of targeted mRNAs, although a few reports have suggested roles in stabilization of target tran-scripts [31]. For example, in humans, Pum1 acts as a negative regulator of LGP2, which is a master activator of innate immunity genes [32]. In *Saccharomyces cerevisiae*, Puf3 protein spe-cifically promotes the degradation of *COX17* mRNA, and Puf5 (Mpt5) protein negatively regu-lates synthesis of HO endonuclease by binding to the 3'-UTR of *HO* mRNA [33]. Yeast Puf6p binds to the *Ash1p* 3′-UTR and downregulates level of *Ash1p* mRNA [34]. PfPuf2 regulates sexual development and sex differentiation in the malaria parasite *Plasmodium falciparum* [35]. Generally, classical Puf proteins have eight Pumilio repeats organized into a crescent-shaped structure that bind to the 3′-UTRs of mRNAs. Each Pumilio repeat contains a tripartite recognition motif (TRM) and generally recognizes a single base of the target RNA via stacking interactions and hydrogen bonding [31]. Hence, upon identification of the TRMs in each Pumilio repeat, it is possible to predict the specific RNA bases to which a Puf protein binds.

In the genome of *P. ultimum*, four genes encode Puf proteins, but their biological functions remain uninvestigated. In this study, we successfully developed a CRISPR/Cas9 system-medi-ated gene knockout and *in situ* complementation in *P. ultimum*. We characterized *PuPuf1* (named *PuM90*), which exhibited increased expression levels specifically during the stage of oospore development in *P. ultimum*. We revealed that *PuM90* is required for *P. ultimum* oospore formation and further identified a flavodoxin-like protein (designated PuFLP) regu-lated by PuM90 through specific binding to the *PuFLP* 3′-UTR. Our findings reveal the mecha-nism through which the RNA-binding protein PuM90 acts as a major regulator of *P. ultimum* oospore formation by repressing *PuFLP* mRNA level. This study represents the first report of both a CRISPR/Cas9-mediated gene editing system in *Pythium* and a post-transcriptional reg-ulatory mechanism for oomycete sexual reproduction.

## Results

### *PuM90* is specifically transcribed during *P. ultimum* oospore formation

Through Hidden Markov Model (HMM) and Basic Local Alignment Search Tool (BLAST) searches, we identified four genes (designated *PuPuf1–PuPuf4*) encoding Puf family RNA-binding proteins in *P. ultimum* (S1 Table), in which four to eight Pumilio repeats were pre-dicted (Pfam ID: PF00806; Fig 1A). Each *P. ultimum* Puf protein exhibited a 1:1 orthologous relationship with those identified in other oomycetes (S1 Fig). Similar to its orthologs, PuPuf1 contained eight predicted Pumilio repeats (Fig 1A and S1 Table). PuPuf1 was designated PuM90 due to the given name of its ortholog PiM90 in *Phytophthora infestans* [36,37]. Based on RNA sequencing analysis (RNA-seq) of *P. ultimum* mycelia cultured in V8 liquid medium

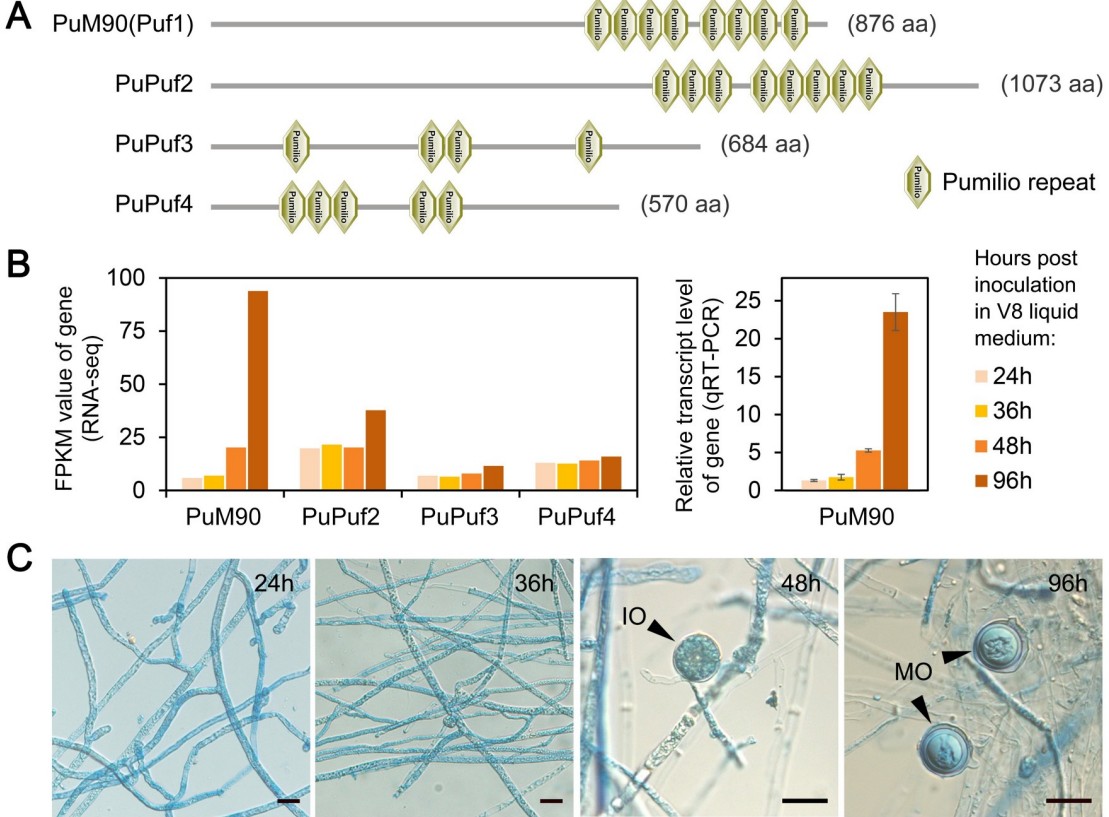

**Fig 1. Characteristics of the Puf family in *P. ultimum*.** (**A**) Predicted Pumilio repeats in Puf proteins. (**B, C**) Transcript levels of the *Puf* genes measured using RNA-seq (Left) and/or qRT-PCR (Right) (B), and sexual development of *P. ultimum* (C) when mycelia were inoculated in V8 liquid medium for 24, 36, 48, or 96 h. Cultures were stained with the lactophenol-trypan blue for 30 s before microscope observation. "IO" and "MO" indicate the oogonia with immature oospore and mature thick-walled oospore, respectively. Bar, 20 μm.

for 24, 36, 48, or 96 h, we found that the transcript level of *PuM90* was strongly increased at 96 h (16.0-fold compared to 24 h; Fig 1B). The transcription pattern of *PuM90* was confirmed through a quantitative reverse transcription polymerase chain reaction (qRT-PCR) assay (Fig 1B). According to our observations, immature oospores appeared at 48 h and mature thick-walled oospores appeared at 96 h (Fig 1C); therefore, the formation of mature oospores coincides with the increase in *PuM90* transcript levels.

### *PuM90* gene knockout and *in situ* complementation in *P. ultimum*

Using a CRISPR/Cas9-mediated gene replacement (with the *hph* gene encoding hygromycin B phosphotransferase) strategy for *PuM90* gene knockout (Fig 2A), we obtained 100 primary transformants and identified 12 candidate mutants through screening based on genomic DNA (gDNA) PCR and sequencing. Three of these candidate mutants were randomly selected for further purification using the single mycelial fragment isolation method. Six single mycelial lines were isolated from each transformant, yielding 18 pure lines in total. Using gDNA PCR with sequencing for confirmation, all pure lines showed homogeneous sequence profiles (Fig 2A), indicating that they carried the same *PuM90* knockout on both alleles of the diploid genome.

An *in situ* gene complementation assay was conducted according to a previously described protocol for *Phytophthora sojae* [38]. After culturing *PuM90* knockout mutants for at least 10

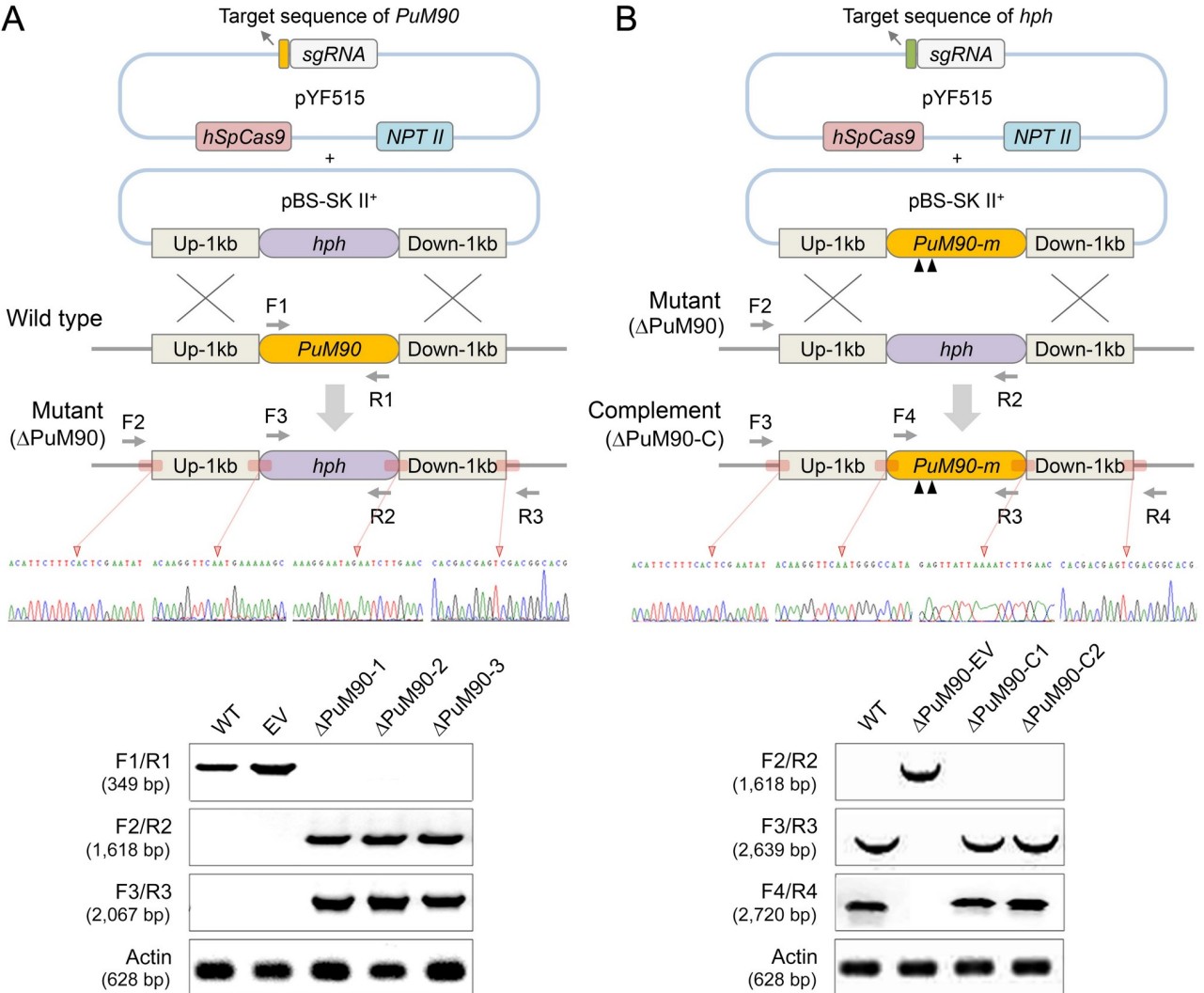

**Fig 2. CRISPR/Cas9-mediated *PuM90* gene knockout and complementation.** (A) Schematic diagram of homology-directed repair-mediated modification of the target gene. Top: an 'all-in-one' plasmid (pYF515) harboring both Cas9 and sgRNA cassettes was co-transformed with a plasmid (pBS-SK II⁺) containing homologous donor DNA *hph* with *PuM90* flanking sequences. Locations of the primers used to screen the HRR mutants and Sanger sequencing traces of junction regions confirming that the *PuM90* ORF was precisely replaced. Bottom: analysis of genomic DNA from the wild-type (WT), empty-vector control line (EV), and *PuM90*-knockout mutants (ΔPuM90-1/2/3) using the primers shown at the top and actin primers as a positive control. **(B)** Schematic representation of the ΔPuM90 mutant complementation strategy and the plasmids used for second transformation in *Pythium*. Top: *PuM90-m* with two black triangles indicates *PuM90* modified with two sgRNA targeting sequences. Locations of the primers used to screen the complementation mutants and Sanger sequencing traces of junction regions confirming that the *PuM90* ORF was precisely complemented. Bottom: analysis of genomic DNA from the wild-type (WT), complemented transformants (ΔPuM90-C1/2), and empty control line of ΔPuM90 (ΔPuM90-EV) using the primers shown at the top and actin primers as a positive control.

generations on antibiotic-free plates, the mutants could no longer grow on G418 antibiotic plates. To avoid persistent cleavage of the newly transformed *PuM90* gene from the previous RNA template, the open reading frame (ORF) sequence of *PuM90* with two mutated single guide RNA (sgRNA) sites (*PuM90-m*) was exploited as donor DNA in the complementation system. This mutated donor DNA vector, along with the *Cas9*-expressing vector and new RNA templates targeting the *hph* gene, was co-transformed into the *P. ultimum* knockout line (ΔPuM90-1 in this study) that had lost G418 resistance. After screening based on gDNA PCR

and sequencing, 10 independent *PuM90-m* complemented strains from 86 primary transformants were identified (Fig 2B and S3 Table).

In this study, we present the results obtained from three representative *PuM90* knockout mutants (ΔPuM90-1/2/3) and two representative *PuM90* complemented strains (ΔPuM90-C1/2). The wild-type strain (WT) and two representative empty-vector lines (EV, in which the *PuM90* knockout was not successful, for comparison with the WT; and ΔPuM90-EV, in which *PuM90-m* was not successfully complemented, for comparison with ΔPuM90-1) were included as controls (Fig 2).

### *PuM90* knockout disrupts oospore formation in *P. ultimum*

After 14 days of growth on V8 agar medium, all three tested *PuM90* knockout mutants (ΔPuM90-1/2/3) and ΔPuM90-EV generated abnormal oospores exhibiting thinner oospore walls (termed abnormal-1 type) or empty oogonia rather than normal oospores (termed abnormal-2 type), while few abnormal oospores were found in the WT, EV, and ΔPuM90-C1/2 lines (Fig 3A and 3B). In addition, no significant differences in the total number of oogonia were observed (Fig 3B), indicating that *PuM90* might be associated with oospore development rather than oogonia development. With ΔPuM90-1/2/3 and ΔPuM90-EV, abnormal oospores were observed not only in the culture medium but also in infected soybean hypocotyl tissue (Fig 3A). In both cases, we found that abnormal-1 type oospores were larger than normal oospores. The average oospore diameters were 22.9–23.9 μm in ΔPuM90-1/2/3 and ΔPuM90-EV, which were significantly larger than 14.9–15.9 μm in the WT, EV, and ΔPuM90-C1/2 cultures (Fig 3A and 3C).

To explore the effects of *PuM90* knockout on oospore formation, oospore morphology was investigated when mycelia were inoculated in V8 liquid medium for 2, 4, and 7 days. No difference was apparent in the 2-day-old oogonia of different strains. However, about half of the 4-day-old oogonia of the *PuM90* knockout mutants showed decreased amounts of cytoplasm in oogonia compared with the WT and EV. Moreover, the cytoplasm of 7-day-old oogonia of the *PuM90* knockout mutants was completely void and empty, whereas the majority of oogonia of the WT and EV at the same stage contained mature thick-walled oospores (Fig 4A). In ΔPuM90-1/2/3 and ΔPuM90-EV, we observed that the thickness of the oospore wall was 0.29–0.36 μm, which was reduced by almost 10 fold compared to the WT, EV, and ΔPuM90-C1/2 (2.01–2.28 μm) (Fig 3D). The difference in oospore wall thickness was confirmed through observations using a transmission electron microscope (TEM) (Fig 4B). The thickness of abnormal-1 type oospores was significantly less in ΔPuM90-1/2/3 and ΔPuM90-EV, and therefore the oogonia were more easily dyed when exposed to 1000 ppm Congo Red solution for 24 h (Fig 4C).

Compared to the WT, EV, and ΔPuM90-C1/2, ΔPuM90-1/2/3 and ΔPuM90-EV showed no significant differences in mycelial growth rate on either nutrient-rich V8 medium (S2A and S2B Fig) or nutrient-poor Plich medium (S2A and S2C Fig), indicating that *PuM90* knockout had no effect on vegetative growth of *P. ultimum*. All hypocotyls of etiolated soybean seedlings inoculated with mycelia of ΔPuM90-1/2/3 or ΔPuM90-EV developed disease symptoms, with lesion sizes (S2A Fig) and relative biomass of *P. ultimum* (S2D Fig) comparable to the WT, EV, and ΔPuM90-C1/2 at 24 h after infection. These results suggest that PuM90 may function specifically in oospore formation in *P. ultimum*.

### Mutations of key amino acids in the TRM of the Puf domain compromise PuM90 function

PuM90 protein contains a typical Puf RNA-binding domain which consists of eight Pumilio repeats and was predicted to organize into a typical crescent-shaped structure (Figs 1A and

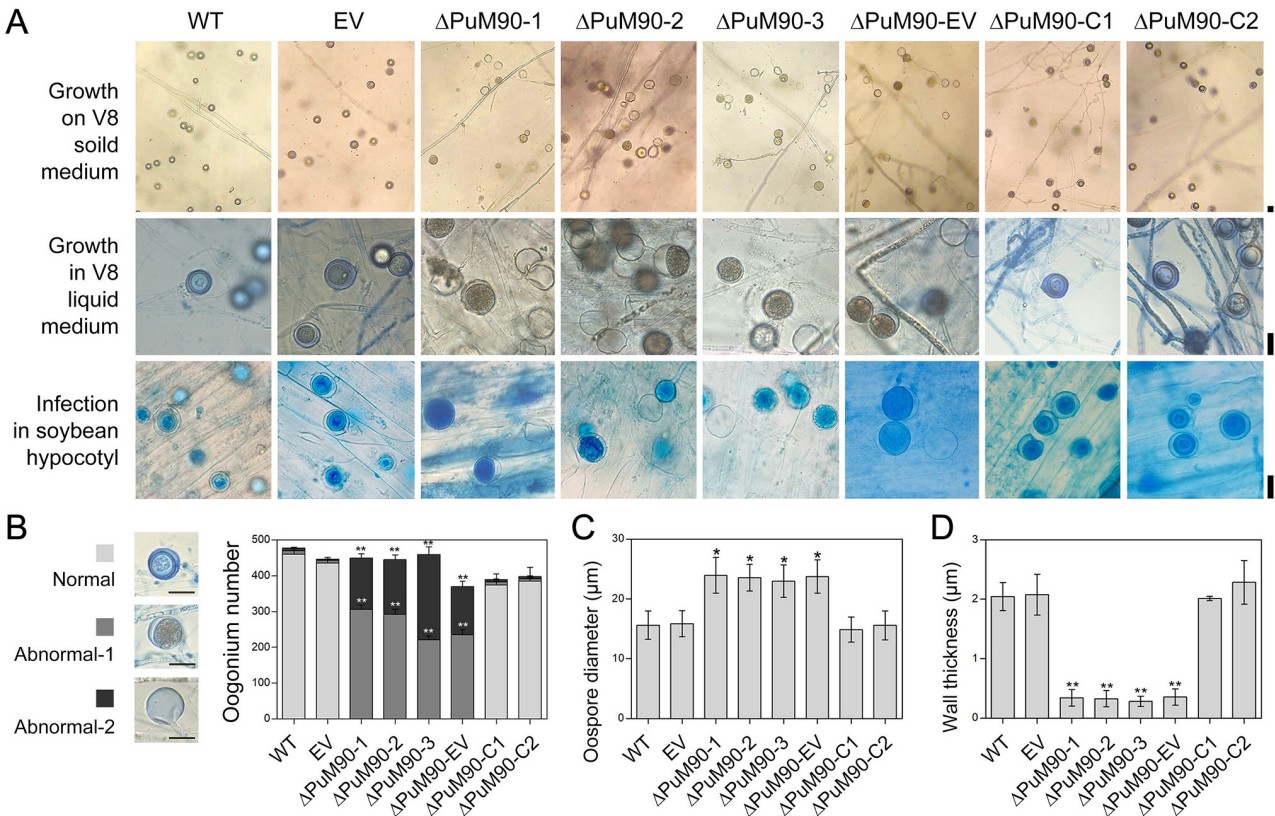

**Fig 3. *PuM90*-knockout disrupts oospore formation. (A)** Morphology of oogonia and oospores generated by WT, EV, *PuM90*-knockout transformants (ΔPuM90-1/2/3), complemented transformants (ΔPuM90-C1/2), and the empty control line of ΔPuM90 (ΔPuM90-EV). Oospores were generated on 14-day-old V8 solid medium (first row), 7-day-old V8 liquid medium (second row), or infected root tissue at 72 hpi (third row) and stained with lactophenol-trypan blue. **(B–D)** Statistical analysis of oogonium number from 14-day-old cultures on V8 solid medium (B), oospore diameter (C), and thickness of the oospore wall (D) from 7-day-old cultures grown in V8 liquid medium. Bar, 20 μm. Asterisks indicate significant differences comparing with WT at P < 0.05 (*) or P < 0.01 (**).

5A). According to a sequence alignment of the Pumilio repeats and reports in yeast [39], we predicted the key amino acids in the TRM that may contact specific targeted RNA bases (Fig 5A). Using an *in situ* complementation strategy, we generated *P. ultimum* mutants that had all 24 amino acids (3 amino acids per repeat × 8 repeats) in PuM90 substituted with alanine (ΔPuM90-CM1/2/3) (Fig 5A and S3 Fig). Similar to ΔPuM90-1/2/3, almost all oospores generated by ΔPuM90-CM1/2/3 were abnormal, including types abnormal-1 and abnormal-2, whereas ΔPuM90-C1 generated normal oospores matching those of the WT (Fig 5B–5E), revealing that mutations of all 24 key amino acids in eight Pumilio repeat compromise the biological functions of PuM90.

## The Puf domain of PuM90 binds to the 3′-UTR of candidate target mRNAs

Interaction of the TRM in the Pumilio repeat with specific RNA bases can result in degradation or instability of the targeted transcripts [40–42]. According to the reported RNA-binding specificity of the TRM [39], we predicted that PuM90 might bind to 3′-UTRs containing a UGUA[A/U/C]AUA motif (Fig 5A). A total of 117 *P. ultimum* genes contained this motif in the predicted 3′ UTR region (within 100 nt after the stop codon). We performed RNA-seq when mycelia were cultured in V8 liquid medium for 24 h and 96 h, and found that there were

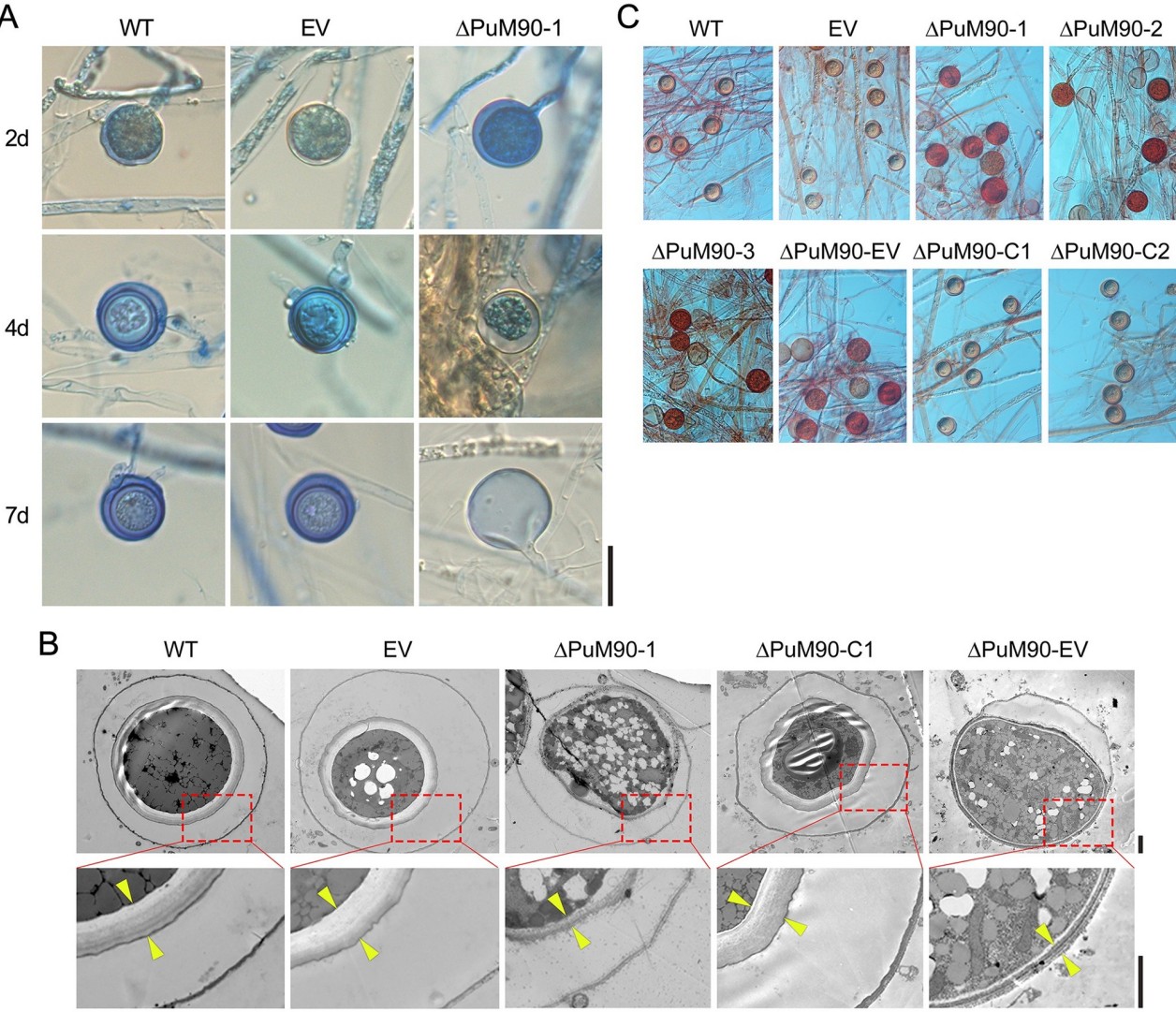

**Fig 4. *PuM90*-knockout transformants produce abnormal oospores. (A)** Oospores from 2-, 4-, and 7-day-old cultures stained with lactophenol-trypan blue were observed under a light microscope. Bar, 20 μm. **(B)** Oospores from 7-day-old cultures were observed with a transmission electron microscope (TEM). The area between the two yellow arrows represents the oospore wall. Bar, 2 μm. **(C)** Oospores from 7-day-old cultures were observed with a light microscope after staining with 1000 ppm Congo red for 24 h. Bar, 20 μm.

more genes differentially expressed at the 96 h (986 genes) compared to the 24 h (80 genes) between ΔPuM90-1 and WT (S2 Table). Among the 363 genes that were significantly upregulated (compared to WT) at 96 h, we identified three candidate PuM90 target genes containing a UGUA[A/U/C]AUA motif in the 3′-UTR (*PYU1_T006554*, *PYU1_T013662*, and *PYU1_T003505*; Fig 6A and 6B). qRT-PCR analysis confirmed upregulation of the three candidate genes at 96 h (Fig 6B). The *PYU1_T013662*-encoded protein contains a flavodoxin-2 domain (Pfam ID: PF02525), and therefore was named Flavodoxin-Like Protein (PuFLP).

Using the electrophoretic mobility shift assay (EMSA), we found that the RNA-binding domain of PuM90 (PuM90-RBD) was sufficient for binding to the selected 30-nt-long 3′-UTR binding sites (3′BSs; upstream 11 nt + UGUANAUA motif + downstream 11 nt) of all three candidate genes (Fig 6C–6E and S4A and S4B Fig). When all of the key amino acids in the

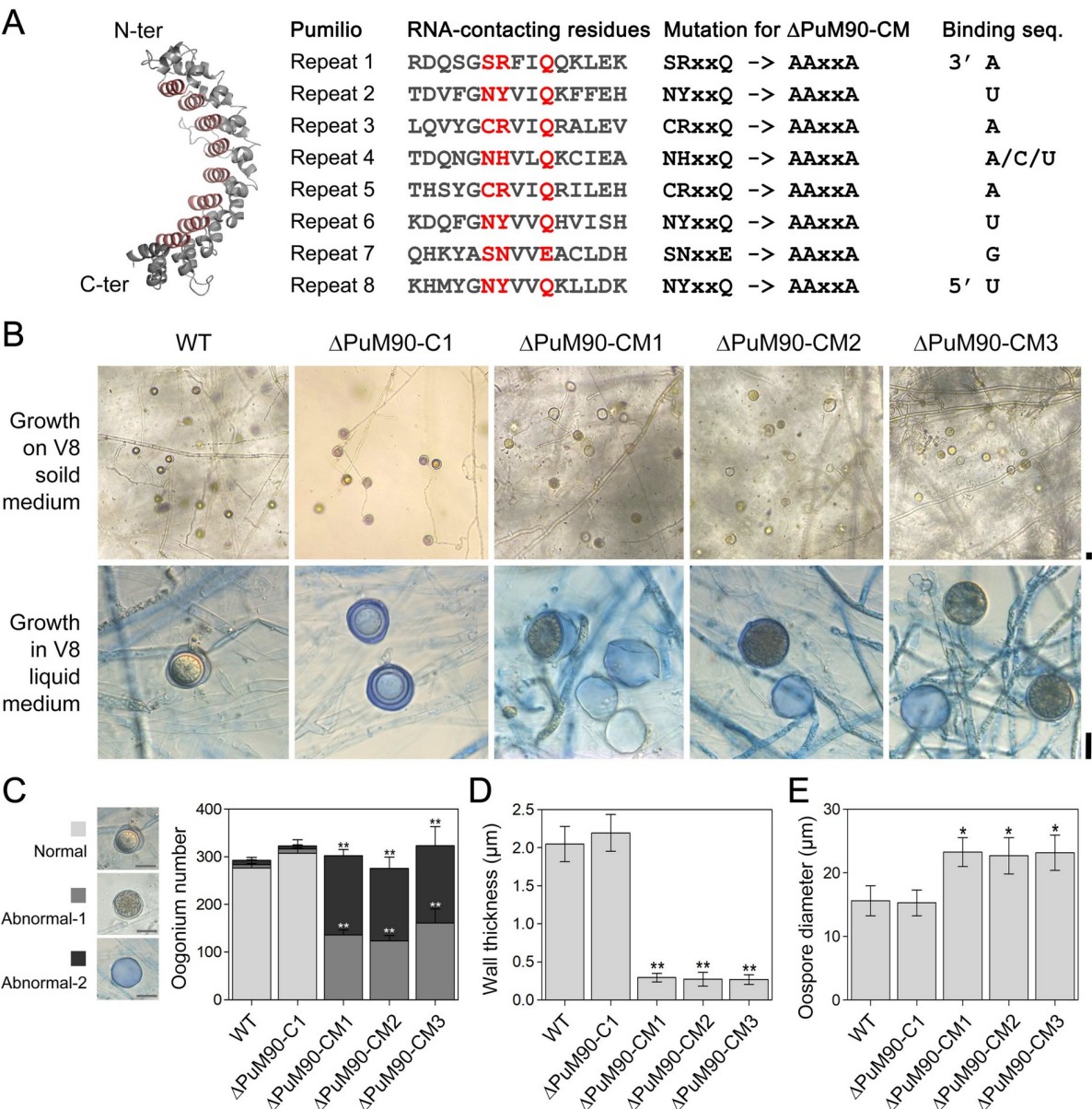

**Fig 5. Mutations of key amino acids in the TRM of the Puf domain compromise PuM90 function. (A)** Protein structure of the Puf RNA-binding domain in PuM90 was predicted using SWISS-MODEL (https://swissmodel.expasy.org/). Individual amino acids predicted to interact with RNA are colored in red and mutated to AAA in ΔPuM90-CM. The predicted RNA bases targeted by each Pumilio repeat are shown. **(B)** Morphology of oogonia and oospores from the WT, *PuM90*-complemented transformants (ΔPuM90-C1), and key amino acid mutation transformants (ΔPuM90-CM1/2/3). Oospores were generated on 14-day-old V8 solid medium (top) or in 7-day-old V8 liquid medium (bottom) cultures. **(C–E)** Statistical analysis of oogonium number from 14-day-old cultures on V8 solid medium (C), thickness of oospore wall (D), and oospore diameter (E) from 7-day-old cultures on V8 liquid medium. Bar, 20 μm. Asterisks indicate significant differences comparing with WT at P < 0.05 (*) or P < 0.01 (**).

TRM of the Pumilio repeat were mutated, the peptide PuM90-RBD$^{AAA}$ exhibited no detectable binding to the 3′BSs (Fig 6D and 6E and S4A and S4B Fig). PuFLP-3′BS formed a weak RNA–protein complex only at a high concentration of PuM90-RBD$^{AAA}$ (Fig 6D and S5A Fig). These data suggest that PuM90-RBD directly binds to the predicted 3′-UTR binding site of three

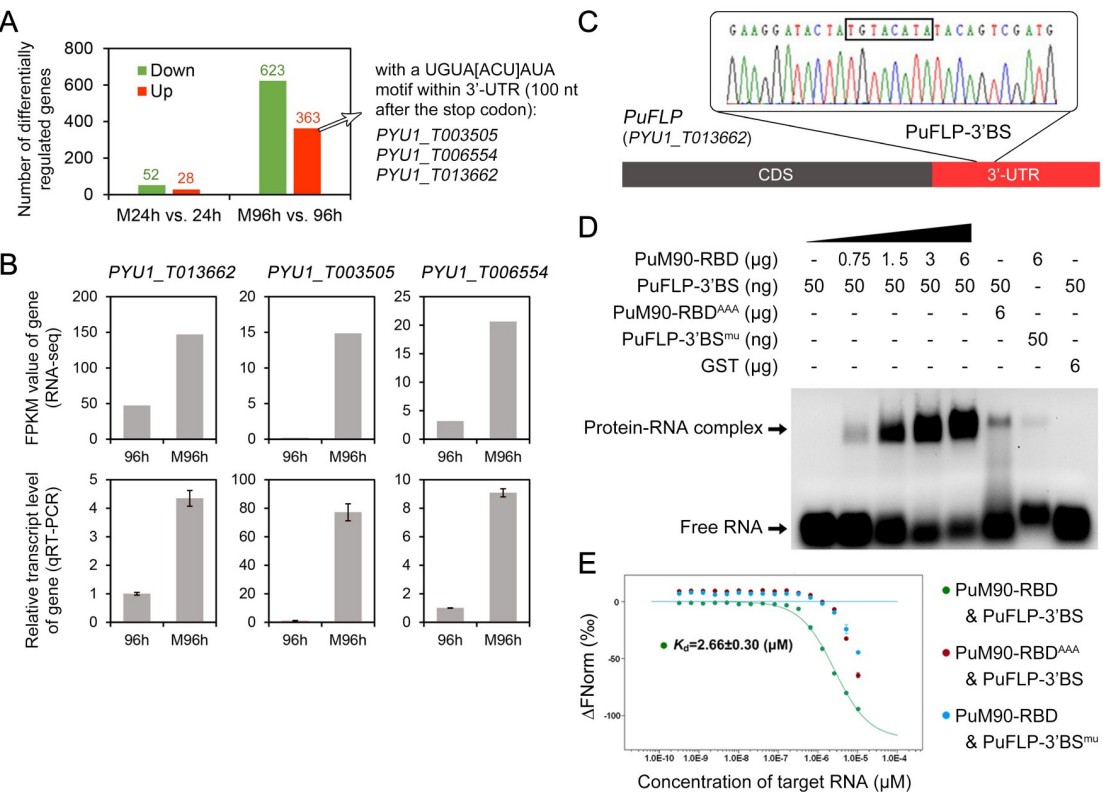

**Fig 6. PuM90 binds to the 3′-UTR of PuFLP and represses *PuFLP* expression. (A)** The number of differentially regulated genes between the *PuM90* knockout mutant ΔPuM90-1 and WT at 24 h and 96 h. Three genes (*PYU1_T006554*, *PYU1_T013662*, and *PYU1_T003505*) were upregulated in the mutants compared to the WT at 96 h and also contained the UGUA[A/U/C]AUA motif in their 3′-UTRs. **(B)** Transcript levels of the three candidate genes measured through RNA-seq and qRT-PCR. **(C)** Sanger sequencing trace of the 30-nt *PuFLP*-3′BS used for the electrophoretic mobility shift assay (EMSA) assay. The UGUACAUA core binding motif is indicated with a black box. **(D)** EMSA results showing that PuM90-RBD bound to PuFLP-3′BS, while PuM90-RBD[AAA] or GST did not bind to PuFLP-3′BS, and PuM90-RBD did not bind to PuFLP-3′BS[mu]. **(E)** Microscale thermophoresis (MST) results showing that PuM90-RBD bound to PuFLP-3′BS ($K_d$ = 2.66 μM, green trace), while PuM90-RBD[AAA] did not bind to PuFLP 3′BS ($K_d$ = 0 μM, red trace), and PuM90-RBD did not bind to PuFLP-3′BS[mu] ($K_d$ = 0 μM, blue trace).

candidate targets and that key amino acids in the TRM of PuM90-RBD were required for this interaction.

## Identification of the flavodoxin-like protein PuFLP as a major PuM90 target

To determine the biological functions of the candidate targets, gene overexpression assays were conducted using the constitutively active *Ham34* promoter. We obtained no transformants exhibiting overexpression of *PYU1_T006554*, and we therefore deleted *PYU1_T006554*. Compared with the WT and corresponding empty-vector control, no significant phenotypic difference was observed when *PYU1_T006554* was knocked out (S6 Fig) or *PYU1_T003505* was overexpressed (S7 Fig; assuming that the GFP tag was not affecting the function of the protein); however, overexpression of *PuFLP* affected oospore formation.

Three representative lines (OE-PuFLP-1/2/3), which showed 9-, 10- and 19-fold increases of *PuFLP* transcript levels (compared to the WT and empty-vector control OE-PuFLP-EV), were selected for phenotypic analysis (Fig 7A). PuFLP protein levels in OE-PuFLP-1/2/3 were measured by Western blot (Fig 7A). Similar to *PuM90* knockout mutants, OE-PuFLP-3

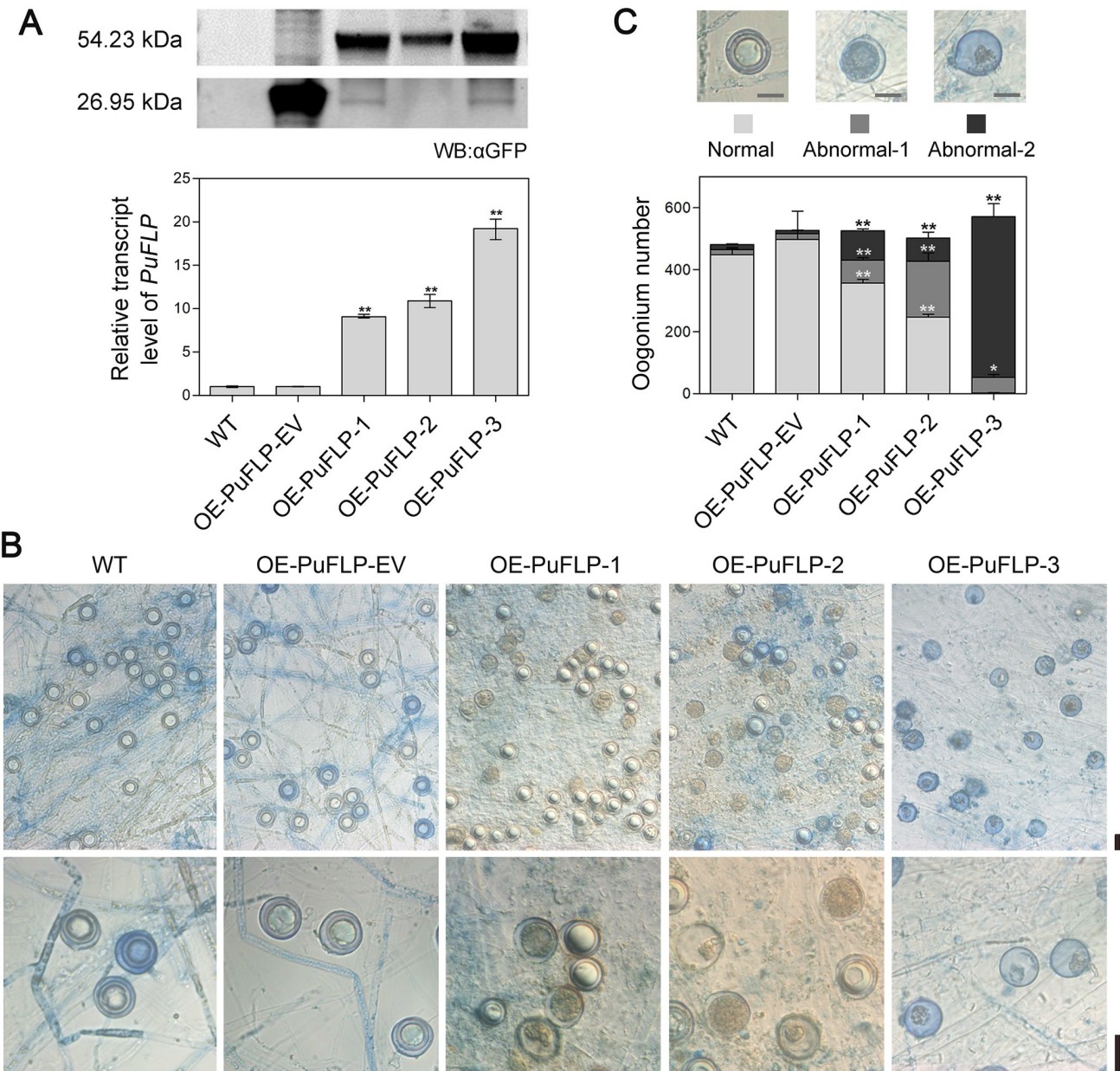

**Fig 7. *PuFLP* overexpression affects oospore development. (A)** *PuFLP* transcript and protein levels measured via qRT-PCR and Western blotting, respectively. Total protein was extracted from each strain for SDS-PAGE and immunoblotting was performed with GFP antibodies. Relative transcript levels were calculated using the WT as a reference. The experiments were repeated independently three times. **(B)** Morphology of oogonia and oospores generated in 14-day-old cultures of WT, an empty-vector control (OE-PuFLP-EV), and *PuFLP* overexpression transformants (OE-PuFLP-1/2/3). **(C)** Statistical analysis of oogonium numbers generated in 14-day-old cultures. Bar, 20 μm. Asterisks indicate significant differences comparing with WT at $P < 0.05$ (*) or $P < 0.01$ (**).

generated abnormal oospores of types abnormal-1 and abnormal-2 and almost no normal oospores. Likely due to their lower overexpression levels, OE-PuFLP-1 and OE-PuFLP-2 generated fewer abnormal oospores (abnormal-1: 13% and 33%; abnormal-2: 17% and 13%) than OE-PuFLP-3, but significantly more than the WT and OE-PuFLP-EV (Fig 7B and 7C). In addition, the transcript level of *PuFLP* was increased at 36 h and 48 h and then repressed at 96

h when *PuM90* was most strongly induced (S8 Fig). These results indicate that PuFLP in *P. ultimum* may play a negative regulatory role during the late stage of oospore formation.

## Mutation of the 3′-UTR binding site of PuFLP affects oospore development

Microscale thermophoresis (MST) experiments revealed that PuM90-RBD bound to PuFLP-3′BS with $K_d$ = 2.66 μM (Fig 6E). Replacement of the UGUACAUA motif in PuFLP-3′BS with ACACACAC (PuFLP-3′BS^mu) abolished its binding to PuM90-RBD (Fig 6D and 6E and S5B Fig), indicating that the UGUACAUA motif of the PuFLP 3′-UTR was essential for PuM90 binding. Using an *in situ* complementation strategy (S9 Fig), we generated *P. ultimum* mutants in which the UGUACAUA motif in PuFLP-3′BS was substituted with ACACACAC (PuFLP^3′BSmu) and used the *PuFLP*-complemented transformants (ΔPuFLP-C) as control. Similar to the phenotypes of *PuM90* knockout mutants, PuFLP^3′BSmu generated large amounts of abnormal oospores including types abnormal-1 (21%) and abnormal-2 (13%), which were significantly higher levels than in the WT (abnormal-1: 4%; abnormal-2: 3%), whereas ΔPuFLP-C generated normal oospores consistent with the WT (Fig 8A and 8B). In PuFLP^3′BSmu, transcript levels of *PuFLP* were 5-fold higher than in the WT and ΔPuFLP-C (Fig 8C). These results indicate that when the UGUACAUA motif in its 3′-UTR was mutated, PuM90 lost the ability to negatively regulate *PuFLP*, resulting in increased *PuFLP* transcript levels that affected oospore formation.

## Discussion

Sexual reproduction is a key process in the pathogenic life cycle of oomycetes, especially those in which asexual reproduction does not occur [43]. However, due to the lack of genome

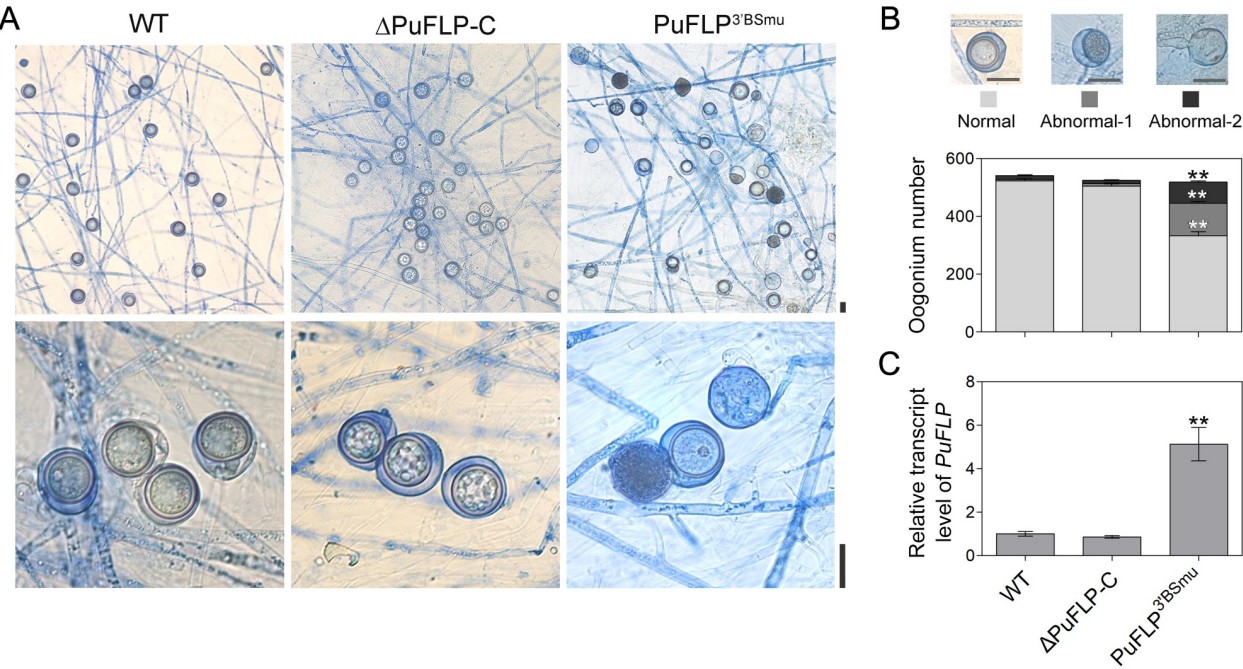

**Fig 8. Replacement of the UGUACAUA motif in PuFLP-3′BS with ACACACAC (PuFLP^3′BSmu) affects oospore development. (A)** Morphology of oogonia and oospores generated in 14-day-old cultures of the WT, *PuM90*-complemented transformant ΔPuFLP-C, and PuFLP-3′BS mutation transformant PuFLP^3′BSmu. **(B)** Statistical analysis of oogonium number generated in 14-day-old cultures. Bar, 20 μm. **(C)** Relative transcript levels of *PuFLP* were analyzed by qRT-PCR, using the WT results as a reference. The experiments were repeated independently three times. Asterisks (**) indicate significant differences comparing with WT at P < 0.01.

editing technologies and functional genomic studies on oomycetes, little is known about the molecular mechanisms of sexual reproduction in oomycetes. Previous transcriptional and functional studies have indicated that M90 is involved in sexual reproduction in the oomycete *Peronophythora litchii*; however, the mechanism through which this RNA-binding protein regulates the biological process remains unknown [43]. In this study, we provide new insights into the functional mechanisms of PuM90 in the regulation of oospore formation in *P. ultimum*. The results demonstrated that at the level of post-transcriptional regulation, a specific interaction of the RNA-binding protein PuM90 with the 3′-UTR of *PuFLP* mRNA is critical for sexual reproduction in *P. ultimum*.

Among the few molecular genetic strategies which have been employed to investigate the biology of *Pythium*, gene silencing via RNA interference (RNAi) is the most commonly applied, e.g., in *Pythium oligandrum* and *Pythium guiyangense* [23,44]. Despite these successes, application of the RNAi strategy has several drawbacks. For example, the degree of gene silencing is unpredictable, and some of the transformants may be unstable [17,45]. In 2016, the CRISPR/Cas9 system was successfully applied to *Phytophthora sojae* for the first time [46], and subsequently to other *Phytophthora* species [18,22]; however, this method had not yet been applied to *Pythium*. In this study, we adapted the CRISPR/Cas9-mediated genome editing tool to *P. ultimum* to delineate the biological functions and functional mechanisms of PuM90 (Fig 2). This initial report of gene knockout and *in situ* complementation in *P. ultimum* indicates that this method is broadly applicable to *Pythium* species.

Puf proteins are highly conserved in terms of protein structure, especially that of the RNA-binding domain, which typically contains eight Pumilio repeats [47]. Despite Puf proteins being ubiquitous among eukaryotes, their functions differ widely among organisms, including roles in stem cell maintenance, development, ribosome biogenesis, and human diseases [31]. Our data indicate that *PuM90* expression was elevated during the oospore formation stage (Fig 1), and knockout of *PuM90* resulted in severe impairment of oospore formation, which was recovered with the reintroduction of *PuM90* into the mutant (Figs 3 and 4).

Puf proteins generally bind to the 3′-UTR of single-stranded RNA targets in a sequence-specific manner, destabilizing the target RNA [48]. For example, in *C. elegans*, a cytoplasmic Puf RNA-binding protein (called *fem-3* binding factor or FBF) binds specifically to the regulatory region of the *fem-3* 3′-UTR, repressing *fem-3* expression and mediating the sperm/oocyte transition [49]. It is reasonable to speculate that PuM90 may bind to the 3′-UTR of its target mRNAs. Based on bioinformatics-based prediction of the 3′-UTR binding motif, RNA-seq for identification of differentially expressed genes, and EMSA for exploration of protein–RNA interactions, we found that PuM90 binds to the *PuFLP* 3′-UTR to repress *PuFLP* mRNA level (Fig 6). PuFLP possesses a domain with a flavodoxin-like fold. Flavodoxins are small soluble electron transfer proteins that occur widely among bacteria and some eukaryotes such as oomycetes and metazoans. Flavodoxins participate in various metabolic pathways; in some bacteria, they have been identified as essential proteins and promising therapeutic targets for fighting bacterial infections, including that of *Helicobacter pylori*, the most prevalent human gastric pathogen [50]. In addition to overexpression (Fig 7), knockout of *PuFLP* also results in abnormal oospore development of *P. ultimum*, indicating that the PuFLP pathway may represent a promising target for fungicide development to control diseases caused by oomycete pathogens.

Although PuFLP$^{3'BSmu}$ generated abnormal oospores (Fig 8), the proportion was lower than those of the *PuM90* knockout mutants (Fig 3). Sequences flanking the core consensus motif in the 3′-UTR were important for binding [51]. Replacement of the UGUACAUA motif in PuFLP-3′BS with ACACACAC (PuFLP-3′BS$^{mu}$) inhibited but did not fully prevent its binding to PuM90-RBD (Fig 6D and 6E and S5B Fig). Therefore, the flanking regions of the

PuFLP-3′BS may play a role in binding of the PuM90-RBD. In addition, each Puf protein generally binds to a large set of mRNAs encoding functionally and cytotopically related proteins [52]. In yeast, Puf1p- and Puf2p-associated mRNAs disproportionately encode membrane-associated proteins, while Puf3p specifically binds to mRNAs encoding mitochondrial proteins [53]. In the present study, the 100-nt region after the stop codon was considered the potential 3′-UTR sequence (Fig 6A); however, some genes may be regulated by a 3′-UTR longer than 100 nt. Some studies have shown that Puf proteins can bind to coding sequences (CDSs) or 5′-UTRs. In *Drosophila* motoneurons, Pum can bind directly to mRNA encoding the *Drosophila* voltage-gated sodium channel paralytic (*para*) [54]. In *Cryptococcus neoformans*, Pum1, an ortholog of both *S. cerevisiae* Puf3p and *Drosophila melanogaster* Pumilio, binds exclusively to the consensus-binding element UGUACAUA in the 5′-UTR of its own mRNA to participate in the regulation of hyphal morphogenesis [55]. Thus, PuM90 may employ other affinity mechanisms for binding to other target genes, perhaps via other *cis* elements in the CDSs or 5′-UTRs.

A typical Puf protein has a C-terminal RNA-binding domain comprised of eight tandem Pumilio repeats along with N- and C-terminal flanking regions. The eight Pumilio repeats are arranged to form an arch, with each repeat composed of three α-helices. The second helix of each repeat contains a TRM, where the residue at position 2 stacks with the cognate base and residues at positions 1 and 5 directly interact with RNA bases through polar interactions. We identified the residues at positions 1, 2, and 5 of the second helix and mutated the corresponding amino acids in all eight repeats. This mutation disrupted RNA binding and the mutants generated abnormal oospores, indicating that PuM90 is a typical Puf protein, and that its ability to specifically bind to target RNA is responsible for its role (Fig 5). However, mutations of 24 amino acids may cause the collapse of the protein structure, resulting in disrupted RNA binding and generating abnormal oospores, thus further researches are needed to ensure structural stability of the protein. By contrast, the N-terminal and central regions differ among Puf proteins. Puf proteins usually collaborate with protein partners to regulate the expression of target mRNA(s). Puf proteins from *Drosophila* and *C. elegans* must interact with other regulatory proteins, such as Nanos and Brat, to function [52]. Three unique domains in the N-terminus of *Drosophila* Pumilio possess repressive activity and can function autonomously [56]. Although there were no predicted functional domains, we speculate that the N-terminal and central regions of PuM90 might play important roles in cofactor recruitment or possess repressive activity; further investigation of these roles is needed.

In phylogenetic analysis, the Puf proteins exhibited a high conservation in oomycetes (S1 Fig). Although the M90 proteins in some oomycete plant pathogens have been revealed to have similar protein domain structure, and/or transcription patterns and biological functions during sexual development [36,37,43], further researches are still needed to learn whether the PuM90-mediated functional mechanisms, including the sequence-specific interaction with the 3′-UTR of target genes and the network of regulated genes, are conserved. In addition to M90, the other Puf members (Puf2–Puf4) showed diverse domain structures and transcription patterns (Fig 1B and S1 Table), thus may possess different functions and mechanisms.

In conclusion, we propose a new model of the role of the Puf RNA-binding protein PuM90 in the sexual development of *P. ultimum* (Fig 9). PuM90 acts as a stage-specific post-transcriptional regulator by specifically binding to the 3′-UTR of *PuFLP* and then repressing *PuFLP* mRNA level. We identified the flavodoxin-like protein PuFLP as a major functional factor involved in oomycete sexual reproduction. We developed a CRISPR/Cas9-mediated method for gene knockout and *in situ* complementation in *Pythium*, providing a robust platform for further molecular studies in *Pythium*. This study describes new technologies and data that will help to elucidate sexual reproduction and post-transcriptional regulation in oomycetes.

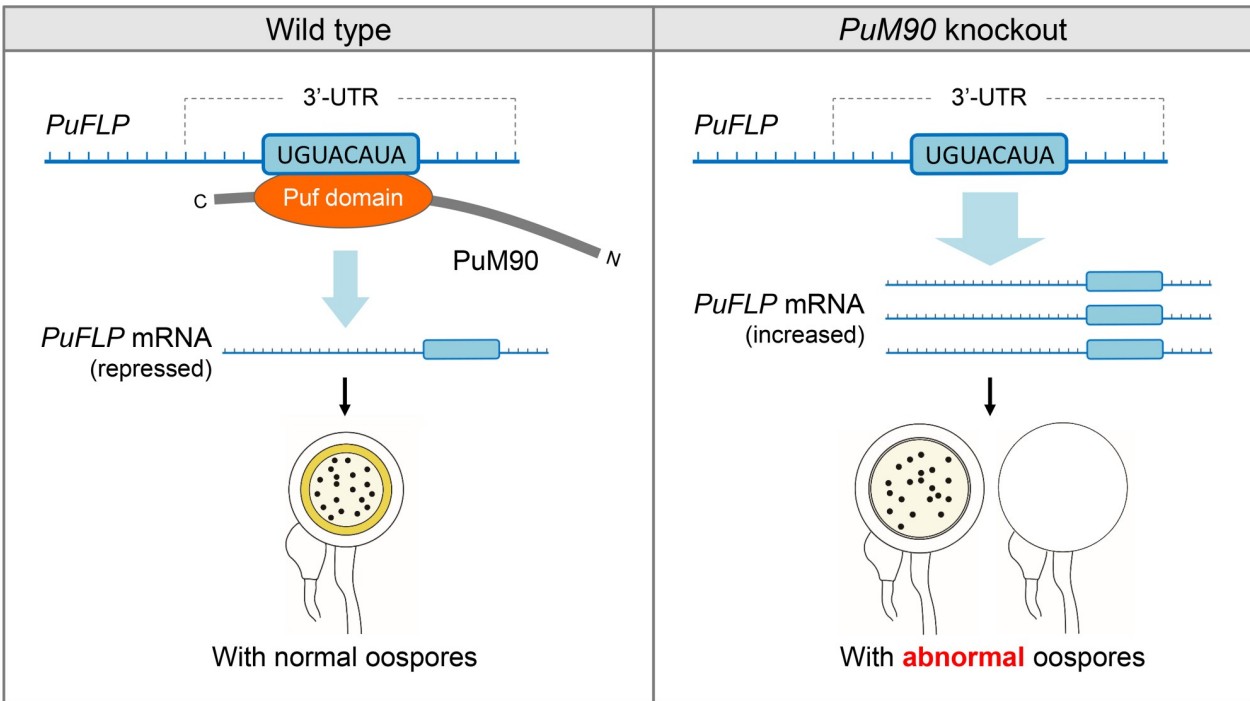

**Fig 9. A proposed model for PuM90 function.** PuM90 protein could specifically bind to a UGUACAUA motif in the 3′-UTR of *PuFLP* mRNA, as the post-transcriptional way to repress *PuFLP* mRNA level to facilitate oospore formation. PuFLP is not regulated by PuM90 in *PuM90* deletion mutants resulting in increased transcript levels of *PuFLP*. Exorbitant *PuFLP* mRNA leads to abnormal oospore. Thus, the RNA-binding protein PuM90 acts as a novel oospore formation regulator to promote the oomycete development.

## Materials and methods

### Source and culturing of *P. ultimum*

The *Pythium ultimum* var. *ultimum* strain F18-6 used as the WT strain was isolated from soybean field soil in Shandong Province, China [57]. All strains employed in this study were routinely grown on 10% V8 agar medium at 25˚C in the dark. For growth rate analysis, the strains were cultured on V8 medium and Plich medium at 25˚C in the dark. Plich medium is a minimal medium with low nutritional content containing 0.5 g $KH_2PO_4$, 0.25 g $MgSO_4 \cdot 7H_2O$, 1 g asparagine, 1 mg thiamine, 0.5 g yeast extract, 10 mg β-sitosterol, 25 g glucose, and 15 g agar per liter [58,59]. The diameter of each colony was measured at 24 hours post inoculation (hpi) on medium, and the average diameter was determined from two measurements taken at right angles to each other. The experiments were repeated three times in triplicate for each assay. Results were compared by *t*-test in Excel.

### RNA-seq sampling and sequencing

RNA-seq samples were collected from mycelia cultured on 10% V8 liquid medium at 25˚C in the dark. For the WT strain, samples were collected at 24, 36, 48, and 96 hpi, and designated 24h, 36h, 48h, and 96h, respectively. Among *PuM90* knockout mutants, ΔPuM90-1 was selected, and samples were collected at 24 and 96 hpi, when *PuM90* expression had not and had been induced, respectively. Three independent biological replicates were conducted for each treatment. RNA was extracted using the EZNA Total RNA Kit I (Omega Bio-tek, Norcross, GA, USA). RNA-seq was conducted by BGI Genomics Co. (Shenzhen, China) using the

BGISEQ-500 system with 100-bp paired-end reads. The filtered clean reads from RNA-seq have been deposited in the National Center for Biotechnology Information (NCBI) database [BioProject ID: PRJNA540115 (Run IDs: SRR15049129-SRR15049146)].

The clean reads were aligned to the genome of *P. ultimum* var. *ultimum* (strain DAOM BR144) with TopHat v2.1.1 (ccb.jhu.edu). A total of two mismatches and gaps per read were allowed, and data were included in further analyses only if both reads in a pair were successfully mapped. Transcript abundance was indicated as fragments per kilobase of exon model per million mapped reads (FPKM). To identify differentially expressed genes, read counts for each gene model were obtained using featureCounts software (bioinf.wehi.edu.au); the fold change and adjusted P-values were calculated using DESeq2 software (www.bioconductor. org); genes with an adjusted P-value < 0.001 and fold change ≥ 2 were considered differentially expressed.

## Transformation of *P. ultimum*

PEG-mediated protoplast transformation was conducted as described by Hua [60] with some modifications to introduce DNA into *P. ultimum*. *P. ultimum* was inoculated into KPYG2 medium containing 1.5 g glucose, 1.0 g yeast extract, 1.0 g peptone, 0.1 g $CaCl_2 \cdot 2H_2O$, 0.02 g cholesterol, 1 g corn oil, and 109.302 g mannitol per liter (with 1.5% agar added for solid medium). After 1 to 2 days (when young hyphae had not grown to the edge of the plates), 40–50 small pieces (3 × 3 mm) from the edge of the mycelial colonies were inoculated into three 250-mL Erlenmeyer flasks containing 50 mL liquid KPYG2 and grown for 2 to 3 days at 25˚C in the dark with shaking once per day. At 2–3 days old, *P. ultimum* mycelial mats (agar plugs need not be excluded) were harvested and pre-treated with 0.8 M mannitol for 10 min, and then digested in 20 mL enzyme solution (0.8 M mannitol; 0.5 M KCl; 0.5 M 2-(N-morpholino) ethanesulfonic acid [MES], pH 5.7; 0.5 M $CaCl_2$; 0.15 g lysing enzymes; and 0.1 g cellulase) for 40 min at room temperature with gentle shaking. The mixture was filtered through a Falcon Nylon Mesh Cell Strainer (BD Biosciences) and protoplasts were pelleted via centrifugation at 1500 g for 4 min in a Beckman Coulter benchtop centrifuge with swing buckets. After washing with 30 mL W5 solution (5 mM KCl, 125 mM $CaCl_2$, 154 mM NaCl, and 177 mM glucose), protoplasts were resuspended in 10 mL W5 solution and held on ice for 30 min. Protoplasts were collected through centrifugation at 1500 g for 3 min in the Beckman Coulter centrifuge and resuspended at $10^6$ cells/mL in MMg solution (0.4 M mannitol, 15 mM MgCl2, and 4 mM MES, pH 5.7).

DNA transformation was conducted in a 50-mL Falcon tube, wherein 1 mL protoplasts was mixed well with 40–50 μg DNA for plasmid transformation. Then, three successive aliquots, each 580 μL, of freshly made PEG solution (40% PEG 4000 v/v, 0.2 M mannitol, and 0.1 M CaCl2) were slowly pipetted into the protoplast suspension and gently mixed. After 20 min of incubation on ice, 10 mL liquid KPYG2 with 50 μg/mL ampicillin was added, and the protoplasts were regenerated at 25˚C in the dark for 15–18 h. For production of stable transformants, the regenerated protoplasts were collected through centrifugation at 2000 g for 5 min in the Beckman Coulter centrifuge, and then resuspended and evenly divided into three Falcon tubes containing 50 mL liquid KPYG2 with 1% agar (42˚C), 30 μg/mL G418 (AG Scientific), and 50 μg/mL ampicillin. The resuspended protoplasts were then poured into empty 90 × 15 mm Petri dishes. Mycelial colonies could be observed after incubation for 48 h at 25˚C in the dark, which were then covered with liquid V8 medium containing 1.5% agar (42˚C), 60 μg/mL G418, and 60 μg/mL ampicillin. Mycelial colonies were observable after 12 h of incubation at 25˚C in the dark. Visible transformants were transferred to 10% V8 agar medium with the addition of 60 μg/mL G418 and propagated for 2–3 d at 25˚C prior to analysis.

Mycelia of *P. ultimum* with high growth rates were spread onto the medium, and the mycelial colonies obtained from the regenerated protoplasts were readily interwoven. Due to the difficulty of zoospore production, we isolated single mycelia to obtain pure transformants [61]. To that end, 5 × 5-mm hyphal plugs were induced on V8 medium plates at 25˚C for 2 days. Then, 2 × 2-cm hyphal plugs taken from near the inoculation points were homogenized with 5 mL water, and 500-μL suspensions were evenly smeared on 2% water agar plates containing 50 μg/mL ampicillin and 60 μg/mL G418, which were incubated at 25˚C in the dark for approximately 24 h. Individual mycelial colonies were checked under a light microscope and transferred to V8 medium plates for subsequent experiments.

## Construction of gene knockout and complementation mutants

Gene deletion mutants were generated using the CRISPR-mediated gene replacement strategy [17]. sgRNAs were designed using the web tool at http://grna.ctegd.uga.edu and listed in S3 Table. Two plasmids, including the "all-in-one" plasmid pYF515 (containing Cas9, sgRNA, and selection marker *NPTII*), and the donor DNA plasmid pBS-SK II⁺ that contained the entire *hph* gene, flanked by 1 kb of homology arms surrounding the *PuM90* gene, were used for PEG-mediated protoplast transformation (Fig 2A). Putative transformants were screened via PCR using the primer combinations shown in Fig 2A. Primer set F1/R1 (S3 Table), which binds within the *PuM90* ORF, was used to screen for deletion of *PuM90* from the genomes of resistant transformants. Primer sets F2/R2 and F3/R3 were used to detect homologous recombination repair (HRR) events (S3 Table). HRR events were analyzed through Sanger sequencing to confirm that *PuM90* was cleanly replaced (Fig 2A). *PYU1_T006554* deletion mutants were constructed using the same strategy (S6A Fig and S3 Table).

For *PuM90* complementation, the knockout mutant was transformed using *NPTII* as the selection marker. The entire gene-coding region with mutated sgRNA sites, which was inserted into two 1.0-kb fragments flanking the target gene, was used as the donor DNA (Fig 2B). The primer set F2/R2 (S3 Table) was used to screen for the deletion of *hph* from the genome of resistant transformants. The primer sets F3/R3 and F4/R4 (S3 Table) were used to detect HRR events. To obtain ΔPuM90-CM (S3 Fig), PuM90-RBD, comprised of three mutated amino acid residues in each of the eight repeats, was synthesized by Genewiz (Suzhou, China). The entire gene-coding region with mutated sgRNA sites was complemented into the *PuM90* knockout mutant, and the resulting transformants were screen as described above (S3 Fig and S3 Table).

To obtain transformants lacking the UGUACAUA motif in PuFLP-3′BS, we attempted to replace the UGUACAUA motif with ACACACAC using gene editing in the WT strain, but failed to obtain any mutants. *PuFLP* deletion mutants were generated using the same CRISPR-mediated gene replacement strategy used for *PuM90* knockout (S9A Fig). The entire *PuFLP* with a mutated UGUACAUA motif and the intact *PuFLP* (with mutated sgRNA sites) as a control were separately complemented into the *PuFLP* knockout mutant (S9B Fig). Putative transformants were screened via PCR using various primer sets and analyzed through Sanger sequencing (S9 Fig and S3 Table).

## Analysis of oospore development

To monitor and quantify oospore production, strains were grown on 10% V8 agar medium at 25˚C in the dark. After 14 days, the cultures were examined via microscopy (Olympus). Three random fields at 40× magnification from each plate were selected for counting of oospores. Diameter and wall thickness of oospores in three random fields were measured using microscope graticules. To allow for explicit observation of the oospores, five 5 × 5-mm hyphal plugs

were cultivated in 8 mL of V8 broth in 90-mm Petri dishes for 7 days at 25˚C in the dark. Oospores stained with the lactophenol-trypan blue (10 mL lactic acid, 10 mL glycerol, 10 g phenol, and 10 mg trypan blue dissolved in 10 mL distilled water) were randomly selected for examination under an inverted microscope (Zeiss) [61]. A TEM was used to observe oogonium sections from the WT, EV, and PuM90 deletion strains. Images were obtained using an AMT camera system. To assess colonization of soybean tissues by oospores, infected epidermal cells were collected at 72 hpi and soaked in lactophenol-trypan blue, and then the infected epidermal cells were examined under an inverted microscope. Oospores from 7-day-old cultures were infiltrated with 1000 ppm Congo red (Shanghai Ryon) and imaged after 24 h to investigate the permeability of the oospore wall. All experiments were performed at least three times independently. Means and standard deviations were calculated using data from three replicates.

## Virulence assay

A virulence assay was performed after hyphal plug inoculation of the hypocotyls of etiolated soybean seedlings of the Williams cultivar, as this cultivar is compatible with *P. ultimum*. Soybeans grown in a greenhouse at 25˚C with a 16-h/8-h light/dark cycle for 4 days were used for hypocotyl infection. Then, hyphal plugs (5 mm in diameter) were inoculated onto the hypocotyls, which were incubated at 25˚C in the dark for 24 h before photography and sampling. Each strain was tested using at least five plants. Virulence was quantified through determination of the ratio of *P. ultimum* DNA to soybean DNA in the infected plants, as measured by qRT-PCR. All assays were repeated independently at least three times.

## Quantitative PCR

Genomic DNA of the *P. ultimum* strains was extracted using a plant DNA kit (Tiangen) from mycelium grown in V8 broth for the gDNA PCR and in infected soybeans for the biomass assay. Total RNA of *P. ultimum* was extracted using the EZNA Total RNA Kit I (Omega). cDNA was synthesized from 1–5 μg total RNA with the PrimeScript First Strand cDNA Synthesis Kit (TaKaRa Bio Inc.) following the manufacturer's protocol. Quantitative PCR was performed in 20-μL reactions containing 20 ng DNA/cDNA, 0.2 mM primers for the target or reference gene, 10 μL SYBR Premix ExTaq (TaKaRa Bio Inc.), and 6.8 μL double-distilled water. PCR was performed on an ABI Prism 7500 Fast Real-Time PCR System (Applied Biosystems Inc.) under the following conditions: 95˚C for 30 s; followed by 40 cycles of 95˚C for 5 s and 60˚C for 34 s; and finally 95˚C for 15 s, 60˚C for 1 min, and 95˚C for 15 s. For the gene expression assay, the actin gene (*PuACTA = PYU1_T009609*) from *P. ultimum* was used as a constitutively expressed endogenous control. To quantify *P. ultimum* biomass in infected soybean tissue, *P. ultimum PuACTA* and soybean *GmCYP2* (*NC_016099.2*) were quantified via quantitative PCR, and the biomass ratio was calculated using the $2^{-[Ct(PuACTA- Ct(GmCYP2)]}$ method [62]. S3 Table lists the primers used in this study. Means and standard deviations were calculated using data from three replicates.

## Western blotting for gene overexpression analysis

To conduct PuFLP-GFP (green fluorescent protein) fusion, the coding region of the *PuFLP* gene was amplified using the primers listed in S3 Table and cloned into the plasmid pTOR:: eGFP [63]. Constructs carrying eGFP-tagged PuFLP and eGFP-tag (control) were transformed into *P. ultimum*. Approximately 150–200 mg of mycelia from stable *P. ultimum* transformants was ground into powder in liquid nitrogen and resuspended in 1 mL 0.01 M phosphate-buffered saline with fresh additions of 1 mM phenylmethylsulfonyl fluoride and 10 μL of protease

inhibitor cocktail (Sigma, Shanghai, China). Total proteins were separated on a 12% sodium dodecyl sulfate polyacrylamide gel electrophoresis (SDS-PAGE) gel and transferred to nitro-cellulose membranes. Anti-GFP antibodies (Abmart Inc., Shanghai, China) were used to detect PuFLP expression (Fig 7A). *PYU1_T003505* overexpression transformants were constructed using the same strategy (S7C Fig and S3 Table).

## Protein expression and purification

Due to the difficulty of expressing and purifying intact PuM90, the CDSs of the PuM90 RNA binding domain (PuM90-RBD) and the PuM90 RNA binding domain with three amino acid residues mutated in all eight repeats (PuM90-RBD[AAA]) were separately inserted into the pGEX-4T-2 vector (containing the glutathione S-transferase [GST] tag; GE Healthcare Life Science) for in vitro assays. The plasmids (GST empty vector, GST-PuM90-RBD, and GST-PuM90-RBD[AAA]) were transformed into *E. coli* strain BL21 (DE3). A 500-mL culture of *E. coli* BL21(DE3) cells was grown at 37°C to an optical density at 600 nm of 0.5, after which gene expression was induced with 0.5 mM isopropyl β-d-1-thiogalactopyranoside (Sigma) for 4 h at 28°C. After lysing of cells, the recombinant proteins were purified on glutathione Sepharose beads (Sangon Biotech, Shanghai, China) and eluted with 5 mM glutathione dissolved in Tris-buffered saline. The concentration of purified proteins was determined using a bicinchoninic acid protein assay kit (Sangon Biotech). Protein purity was assessed via SDS-PAGE.

## Electrophoretic mobility shift assay

EMSA was performed as described previously [64]. 5-Carboxy-fluorescein (FAM)-labeled probes comprised of the core motif of the 3′-UTR of target sequences were synthesized by Genewiz (Suzhou, China). Labeled RNA fragments (50 ng) were mixed and incubated with various concentrations of purified PuM90-RBD protein at 25°C for 30 min in EMSA/Gel-Shift Binding Buffer (Beyotime). The mixtures were then loaded onto a 1% agarose gel and electro-phoresed for 1 h. EMSA signals (labeled RNA fragments) were detected using Alexa 488 with the VersaDoc imaging system (Bio-Rad, Philadelphia, PA, USA).

## Microscale thermophoresis

Binding of PuM90-RBD protein to the 3′-UTR of PuFLP labeled with FAM was detected through an MST assay using the Monolith NT.115 instrument (NanoTemper Technologies), according to a previously described procedure [65]. A constant concentration (10 μM) of the labeled 3′-UTR in MST buffer (50 mM Tris, pH 7.5, 150 mM NaCl, 10 mM MgCl$_2$, and 0.05% Tween 20) was titrated against increasing concentrations of PuM90-RBD protein dissolved in double-distilled water. MST premium-coated capillaries (Monolith NT.115 MO-K005) were used to load the samples into the MST instrument at 25°C using high MST power and 60% light-emitting diode power. Laser on and off times were set to 30 s and 5 s, respectively. All experiments were conducted in triplicate. Data were analyzed using NanoTemper Analysis software v. 1.2.101 (NanoTemper Technologies).

## Supporting information

**S1 Fig. Phylogenetic relationships of the identified Puf proteins in oomycetes.** The phyloge-netic trees were constructed using neighbor-joining method with 1,000 bootstrap replicates in MEGA 7.0 software. Bootstrap values higher than 80 are displayed.
(TIF)

**S2 Fig. *PuM90* is not involved in mycelial growth or virulence. (A)** Growth characteristics at 24 h after inoculation on V8 medium and Plich medium, and virulence on soybean hypocotyls of WT, EV, PuM90-knockout transformants (ΔPuM90-1/2/3), complemented transformants (ΔPuM90-C1/2), and the empty control line of ΔPuM90 (ΔPuM90-EV). **(B, C)** Growth rates on V8 medium (B) and Plich medium (C). **(D)** Relative *P. ultimum* biomass detected through qRT-PCR at 24 h after hypocotyl infection.
(TIF)

**S3 Fig. CRISPR/Cas9-mediated mutation of key amino acids in the RNA-binding domain of PuM90. (A)** Locations of the primers used to screen for complemented transformants (top) and Sanger sequencing traces of the mutated sequence regions in all eight Pumilio repeats (bottom). **(B)** Analysis of genomic DNA using the primers shown in (A) and actin primers as a positive control.
(TIF)

**S4 Fig. Results of EMSA assays for the RNA-binding domain (RBD) of PuM90 and its candidate targets.** EMSA results showing that PuM90-RBD bound to the 3′BS of *PYU1-T006554* (A) and *PYU1-T003505* (B), while the mutant PuM90-RBD$^{AAA}$ and GST did not.
(TIF)

**S5 Fig. Key amino acids in eight Pumilio repeats of PuM90-RBD and the UGUACAUA motif in the PuFLP 3′BS are essential for binding activity.** (A) The PuM90-RBD$^{AAA}$ peptide nearly abolished detectable binding to PuFLP-3′BS. (B) The PuM90-RBD peptide nearly abolished detectable binding to PuFLP-3′BS$^{mu}$ but formed a weak RNA–protein complex at a high PuM90-RBD concentration.
(TIF)

**S6 Fig. *PYU1_T006554*-knockout does not affect oospore formation. (A)** Construction and verification of three representative candidate mutants for *PYU1_T006554* knockout. **(B, C)** Morphology (B) and number (C) of oogonia generated in 14-day-old cultures. Bar, 20 μm.
(TIF)

**S7 Fig. *PYU1_T003505* overexpression does not affect oospore formation. (A, B)** Morphology of oogonia and oospores (A) and number of oogonia (B) generated in 14-day-old cultures. Bar, 20 μm. **(C)** PYU1_T003505 protein levels measured through Western blotting. **(D)** *PYU1_T003505* transcript levels measured through qRT-PCR. Asterisks (**) indicate significant differences comparing with WT at P < 0.01.
(TIF)

**S8 Fig. Transcription analysis of *PuFLP*.** Transcript levels of the *PuFLP* genes measured using RNA-seq (top) and qRT-PCR (bottom) when *P. ultimum* mycelia were cultured in V8 liquid medium for 24, 36, 48, or 96 h.
(TIF)

**S9 Fig. CRISPR/Cas9-mediated PuFLP-3′BS mutation. (A)** Construction and verification (with Sanger sequencing traces) of the *PuFLP*-knockout mutant (ΔPuFLP). **(B)** Construction and verification (with Sanger sequencing traces) of the complementation line PuFLP$^{3'BSmu}$, in which the UGUACAUA motif in PuFLP-3′BS was replaced with ACACACAC. **(C)** Analysis of genomic DNA PCR products using the primers shown in A and B and actin primers as a positive control.
(TIF)

**S1 Table. Puf proteins identified in *Pythium ultimum* and other oomycetes.**
(XLSX)

**S2 Table. Differentially regulated genes when *PuM90* was knocked out.**
(XLSX)

**S3 Table. Primers and sgRNAs used in this study.**
(XLSX)

## Acknowledgments

We would like to thank Dr. Haifeng Zhang, Dr. Yan Wang, Dr. Kaixuan Duan, Dr. Bo Yang, and Dr. Meixiang Zhang (Nanjing Agricultural University, Nanjing) for helpful discussions.

## Author Contributions

**Conceptualization:** Hui Feng, Yuanchao Wang, Xiaobo Zheng, Wenwu Ye.

**Data curation:** Hui Feng, Zhichao Zhang, Maozhu Yin, Wenwu Ye.

**Formal analysis:** Hui Feng, Wenwu Ye.

**Funding acquisition:** Yuanchao Wang, Xiaobo Zheng, Wenwu Ye.

**Investigation:** Hui Feng, Chuanxu Wan, Zhichao Zhang, Han Chen, Zhipeng Li, Haibin Jiang, Maozhu Yin.

**Methodology:** Hui Feng, Zhichao Zhang, Han Chen, Zhipeng Li, Haibin Jiang, Daolong Dou, Yuanchao Wang, Xiaobo Zheng, Wenwu Ye.

**Project administration:** Suomeng Dong, Yuanchao Wang, Xiaobo Zheng, Wenwu Ye.

**Resources:** Daolong Dou, Yuanchao Wang, Xiaobo Zheng, Wenwu Ye.

**Software:** Zhichao Zhang, Maozhu Yin, Wenwu Ye.

**Supervision:** Suomeng Dong, Yuanchao Wang, Xiaobo Zheng, Wenwu Ye.

**Validation:** Chuanxu Wan.

**Visualization:** Hui Feng, Wenwu Ye.

**Writing – original draft:** Hui Feng, Wenwu Ye.

**Writing – review & editing:** Hui Feng, Suomeng Dong, Yuanchao Wang, Xiaobo Zheng, Wenwu Ye.

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
