## [Decision Letter · Decision Letter 0]

6 Sep 2021

Dear Dr. Ye,

Thank you very much for submitting your manuscript "Specific interaction of an RNA-binding protein with the 3′-UTR of its target mRNA is critical to oomycete sexual reproduction" for consideration at PLOS Pathogens. As with all papers reviewed by the journal, your manuscript was reviewed by members of the editorial board and by several independent reviewers. The reviewers appreciated the attention to an important topic, and I especially was pleased to see molecular genetic studies being applied to Pythium.

I do not believe that additional experimentation is required for the paper, even though reviewer #1 has made suggestions for additional experiements. Nevertheless, each reviewer has made important suggestions for improving the manuscript. We are likely to accept this manuscript for publication, providing that you modify the manuscript according to the reviewers' recommendations.

Sincerely,

Howard S. Judelson

Guest Editor

PLOS Pathogens

Bart Thomma

Section Editor

PLOS Pathogens

Kasturi Haldar

Editor-in-Chief

PLOS Pathogens

orcid.org/0000-0001-5065-158X

Michael Malim

Editor-in-Chief

PLOS Pathogens

orcid.org/0000-0002-7699-2064

Reviewer Comments (if any, and for reference):

Reviewer's Responses to Questions

**Part I - Summary**

Reviewer #1: In this manuscript, the authors describe the identification and characterization of RNA binding protein called PuM90. This protein is found within Pythium ultimum and supposedly has orthologs within other oomycetes, such as Phytophthora infestans. The authors describe how PuM90, through its pumilio containing domains, bind the 3' UTR of another gene called PFLP and negatively regulates its expression.

The authors hypothesize that the formation of oospores is regulated through PuM90 activity and its consequent modulation of PFLP expression. Overall, the manuscript is well written and presents high-quality data. I am particularly impressed with the authors' genetic analyses, using CRISPR/CAS9 and complementation assays to validate PuM90 functionality.

The authors demonstrate that PuM90 is a gene induced during oospore formation and that it encodes a protein with pumilio motifs. The authors identify four different pumilio domain-containing proteins, each of which has different number of RBP domains and predicted length.

Unfortunately, the authors do not show what other domains and configurations are present within this protein family (is it a family?). It is an important question as the length and composition of these proteins seems to be very diverse.

Next, the author's look at the transcription level of PuM90, and do so by using RNA-Seq of RNA derived from mycelia. Whilst it's exciting and beneficial to identify genes associated with sporulation, much of the work presented was not done in vivo (during infection).

For example, the authors generated PuM90 mutants and complementation mutants and assessed their impact on sporulation in vitro and vivo. Equivalent in vivo work for PuFLP is not presented.

Given that PuM90 is predicted to be an RNA binding protein, the authors identified possible binding partners or target RNAs. They did so by using the modularity of the PuM90, which allows them to predict the RNA sequence mofit PuM90 would bind. In doing so, and through computational analysis and gene expression analysis, they identified three strong candidates that seemed to be regulated.

The authors then used RNA binding essays to demonstrate that PuM90 can bind to the 3'UTR of these three transcripts and that by removing or modifying PuM90 sequence-specificity, binding is limited or reduced. Unfortunately what the authors did not show whether this was sequence-specific. It would have been informative to mutate the 3' UTR binding motif and see whether that impacts on

binding in these essays. Alternatively, the authors could have selected a UTR for a transcript that they do not expect PuM90 to target.

One critical experiment in demonstrating a direct relationship between PuM90 and PuFPL1 is to assess phenotypes and relationships between mutations. In this regard, the question as to whether overexpression of PuFLP phenocopies PuM90 deletion is important. Indeed, the authors show that overexpression of PuFLP leads to reduction of oospore formation. These results strongly suggest that PuM90 acts on PuFLP transcripts, and by extension, regulates oospore formation. What is somewhat surprising, however, is that deletion of PuFLP returns a wild-type phenotype. Given that this protein inhibits oospore formation, one would expect some sort of a developmental phenotype. It also begs the question: what would a PuM90/PuFLP double mutant phenotypically look like? The actual mechanism(s) may be a little more intricate as the presented model would suggest. If both PuM90 absence and high levels of PuFLP are required for oospore development to be perturbed, one could think that the product(s) arising from PuFLP processing (prompted by PuM90) has a role in oospore formation. A double mutant may clarify this point in the future.

Overall, the data itself and in the way it is presented, is convincing and strongly supports a model suggestive of a new mechanism for gene regulation in Pythium. However, there are several outstanding questions that I think would dramatically increase the impact and importance of this work.

Firstly, the author states that PuM90 is an ortholog of Phytophthora infestans. However, no such evidence is presented. How many M90-like proteins are there in P. infestans and P. ultimum? If there are more than one, are they true orthologs?.

Secondly, the authors suggest and invoke a mechanism that may be conserved across the oomycetes. It would be particularly interesting to understand protein family composition across the oomycetes and to know whether there's any evidence of expansion or deletion in specific lineages. For example, what would regulation look like in pathogens that produced other spore types? Providing a couple of hints, through some rather basic analyses, would raise the interest of this work considerably.

Reviewer #2: The manuscript by Feng et al. reports the identification a Puf family RNA-binding protein, PuM90, as an important player in Pythium ultimum oospore formation, and the underlying post-transcriptional regulatory mechanism. The Puf domain of PuM90 specifically binds to a UGUACAUA motif in the mRNA 3′ untranslated region (UTR) of PuFLP, a flavodoxin-like protein, and thereby downregulate PuFLP expression to facilitate oospore formation. This study not only sheds important insight into the regulation of sexual reproduction of P. ultimum, but also reports CRISPR/Cas9 system-mediated gene knockout and in situ complementation methods for Pythium for the first time. P. ultimum is a homothallic oomycete and sexual reproduction plays an important role in its life cycle. The identification of key components in oospore formation is significant as these components can serve as potential targets of disease control. This study is well designed and well executed. The manuscript is concise and well written overall. However, some parts may need be more detailed to be readily understandable by the readers.

Reviewer #3: This MS focusses on a RNA-binding protein in the oomycete plant pathogen Pythium ultimum that is named named PuPuf1 aka PuM90. The authors nicely show that transformants lacking this protein show defects in oospore formation and they unravel how a domain in PuM90 that carries pumilio repeats targets a short region in the 3’UTR of mRNAs thereby repressing translation of those mRNAs. One of the target genes that is identified is PuFLP. It encodes a flavodoxin-like protein and transformants overexpressing PuFLP are phenocopies of the PuM90 knock-out transformants. The MS is a pleasure to read and the research is technically sound. I have a few questions and comments, and a list of items to need to be considered when revising the MS.

**Part II – Major Issues: Key Experiments Required for Acceptance**

Reviewer #1: Suggested experiments listed above (though they do not stand in the way of publication in my view)

Reviewer #2: (No Response)

Reviewer #3: (No Response)

**Part III – Minor Issues: Editorial and Data Presentation Modifications**

Reviewer #1: Some minor editorial issues (Western blot with a capital 'w').

Reviewer #2: All the primers are listed in Table S3. Although short notes are included under “Application”, in some cases it is difficult to figure out how these primers were used. The authors should add more detailed description in the table or table notes, associate the names better with the main manuscript, differentiate sgRNA target sequences and gene specific sequences from added sequences for cloning purpose. The sgRNA target sequences should be also included in the main manuscript.

For all figures with transcript levels from RNA-seq and RT-qPCR, please describe what the number values represent in the legends. For example, Fig 1B, Fig 6B and more.

Fig. 2, The primers in A and B are very confusing. F2/R2 seems to be the same in both panels. How about others? In Table S3, there seems to be a set from F1/R1 up to F4/R4. However, the primers in both panels do not correspond. The authors need clarify and label them clearly.

For multiple figures showing statistically significant difference, the author should clearly indicate to which it the comparisons were made.

Fig. 4B, describe the red square boxes, add scale bar for the bottom panel.

Line 522, 2 [Ct(PuACTA-Ct(GmCYP2)] should be 2 -[Ct(PuACTA)-Ct(GmCYP2)]. Please double check.

Line 530. Add reference for pTOR::eGFP, or describe it.

For line 191-202, Key amino acids in the TRM of the Puf domain determine PuM90 function. 24 amino acids were mutated to Alanine. This may cause the collapse of the protein structure. In this case, the conclusion is not warranted. To interpret the results more accurately, the authors may include a short discussion on this, and change “Key amino acids in the TRM of the Puf domain determine PuM90 function” to “Mutations in key amino acids in the TRM of the Puf domain compromise PuM90 function”.

Reviewer #3: One aspect that I miss in the discussion is a reflection on the potential activity of the flavodoxin-like protein, and why and how its presence disturbs oospore formation. I conclude from lines 259-260 that PuFLP knock-out transformants were generated but there is no description of the phenotype. Is FLP required for asexual development or for virulence? And what is the overall expression profile of PuFLP?

Relation between PuM90 and PiM90.

Line 122-124: this is the first time in the MS that the P. infestans ‘ortholog’ M90 is mentioned. Is PuPuf1 really the ortholog of PiM90? There are at least two publications on M90 in P. infestans but only one is included as reference. That is #31 (Fabritius et al.) in which one can trace back that M stands for mating and 90 is just a random number of a cDNA clone. The authors should reconsider the name ‘PuM90’. Adapting that name requires data showing that PuPuf1 is really the Puf that is closest to PiM90. So please add a phylogenetic tree of Puf’s in P. ultimum and in Phytophthora species, also to clarify if there are additional Puf’s in Phytophthora apart from M90. Fabritius et al. (31) noted that it is single copy gene based on Southern blot analysis. The other paper on M90 (Cvitanich and Judelson 2003 Eukaryotic Cell) is a more in depth study showing, amongst others, that in P. infestans M90 is also expressed during asexual development. Expression in P. infestans thus differs from expression in Pythium. It is worth to note that, and to include a reference to Cvitanich and Judelson (2003).

In the discussion also PlM90 from Perenophytophthora litchi is mentioned (line 275-276) (reference #37). Also this gene is expressed during asexual reproduction as well as sexual reproduction. How does PlM90 relate to PuPuf1 (PuM90) and PiM90?

Abstract. Revisit line 21 -24. Is it so that binding of PuM90 to the 3’ UTR of the mRNA causes inhibition of translation, thereby resulting in less PuFLP protein? Line 24 says ‘..downregulates PuFLP expression.. ‘ which suggests interference at the level of gene expression, so at the DNA level.

Line 50-53: revisit this sentence. It is hard to read. Also, I find word ‘interestingly’ misplaced and the statement in line 53 (‘...has become more important…’) confusing. It suggests that over time, oospore production has taken over zoospore production. But is that true? Is there evidence and is this described in reference #8? Referring to reference #7 in line 53 is likely wrong. It a review paper on Phytophthora infestans, not on Pythium ultimum. Check.

Line 24: downregulates

Line 28: critical to = critical for

Line 37: is reference 2 appropriate for the statement that ‘all known oomycetes can undergo sexual reproduction’ ? Unlikely giving the title of the paper. Please check.

Line 58: sensitive OR more sensitive?

Line 59: Also, P. sojae PsGK5-silenced mutants lost the ability to produce oospores. Refer to Yang et al. 2013 Molecular Microbiology 88, 382-394.

Line 62-63: In addition? Revisit this sentence. What is meant by ‘critical regulators’? Could PsYPK1 (functions as a kinase) be considered as a critical regulator?

Line 64: Publication? OR is it: ‘Availability’, or maybe ‘release’?

Line 66: protoplast electroporation? For protoplasts PEG-mediated transformation is common. For zoospores, electroporation.

Line 67 and 72: Weiland 2003 Current Genetics showed protoplast transformation of Pythium aphanidermatum.

Line 67: Vijn 2003 FGB showed Agrobacterium mediated transformation of P. ultimum

Line 68: also in P. infestans. Include the reference to the recent paper (2021) of the Judelson lab in MPP.

Line 75-76: revisit this sentence. ‘..during RNA [……] stability….’ Is not correct.

Line 75: I notice that the authors consider ‘regulation of gene expression’ as a process that occurs at all levels. I have learned that gene expression is a process that occurs solely at the DNA level so based on that, gene expression cannot be regulated at the post-transcriptional level and (in line 84) targeted mRNAs cannot be expressed. In this line of thinking proteins cannot be expressed so in line 88 the word ‘expression’ is misplaced and should be replaced by, for example ‘synthesis’ or ‘production’. Line 105-106: PuM90 acts at the RNA level so does it actually ‘repress translation of PuFLP mRNA’ rather than ‘repress expression of’ the gene ?

Line 90: ‘…downregulates translation of Ash1p mRNA.’

Line 96: Is it more precise to write: ‘…it is possible to predict the specific RNA sequence to which a PUF protein binds’ ?

Line 102: was = is

Line 117: ‘… the transcript level of PuPuf1 was strongly increased….’

Line 121: here the authors take a too sharp corner when stating that PuPuf1 may be involved in oospore formation. One could say that ‘this coincides with the increase in PuPuf1 transcript levels’.

Legend figure 1; line 746: …repeats in PUF proteins….

Line 747: Transcript levels NOT transcription levels. PUF genes OR Puf genes as the figure and in the main text? Be consistent.

Figure 1B: rephrase ‘hours post [..] culturing’. That is not correct.

Figure 1C and the legend of Figure 1C: the reader has to guess what is shown here. Point out the gametangia (although I cannot distinguish gametangia in these photo’s), oospores etc. Indicate in the legend how the tissue was stained.

Line 127: improved? Why improved and compared to which strategy?

In figure 2 lower panels: indicate the sizes of the PCR fragments.

Line 762: the triangles are pink colored not black.

Line 139: rephrase ‘..they would be capable of further transformation…’.

Line 146: how many complemented strains were obtained?

Legend Figure 3A; lines 773 and 774: first line = first row etc.

Line 774 and 781: … stained with lactophenol-trypan blue… (rather than ‘treated’)

For the data shown in Figure 3B and 3C it is not clear how oospore diameter and wall thickness, respectively, was determined. How was wall thickness measured in the photographs in Figure 3A? line 177-178 suggests the TEM is confirming the difference in thickness implying that the original measurements were based on bright field images.

Line 161: figure 3B shows total number of oospores, not oogonia. Check.

Line 169: ‘… was investigated in mycelium cultured for 2, 4 and 7 days in V8 broth.’

Line 193-193: revisit this sentence. It reads as if the PuM90 protein was predicted to organize into a typical crescent-shaped structure but this is limited to the domain carrying the repeats.

For Figure 5D and 5E same question as for Figure 3: it is not clear how oospore diameter and wall thickness, respectively, was determined. How was wall thickness measured?

Figure 5A: mutantion = mutation

Line 201 – 202: was there an attempt to distinguish between repeats? Do all repeats have to possess the three essential amino acids for proper functioning of PuM90 in oospore development? Consider to rephrase this sentence. When first reading it I was stuck by ‘THESE three amino acids’ whereas it is actually 3 x 8 amino acids.

Line 208 -210: Suggestion for rephrasing: By minng/analysing the 3’ UTR sequences (i.e. 100 nt after the stop codon) of all predicted (?) P. ultimum genes we identified 117 genes that contain this motif in their 3’ UTR.

Line 213: ‘transcript level’ OR ‘higher expression level’ but NOT transcription level

Line 210-213: revisit this sentence. The formulation is too complicated.

Figure 6A: terminator codon = stop codon

Figure 6B: Y-axis ‘transcript level’ NOT transcription level. Also change this the legend. And check throughout the MS. See for example line 240.

Heading Table S2: … knocked out….

Table S2: Explain what is shown in each column in a text block next to the table or as notes between title and actual table.

Line 215-216: any further information about the other two genes? And why was 13662 chosen for further study and not the other two?

Line 222: binding site (not sites)

Line 241: protein level – delete ‘expression’

Table S1: what is shown in column M24h?

Fig S6: the overexpression of PYU1_T003505 in P. ultimum is accomplished by introduction of a GFP fusion construct. Is it possible that the GFP tag interferes with the function of PYU1_T003505 and that therefore overexpression does not show a change in phenotype?

Line 283: why ‘unique’ ?

Line 285: the drawbacks are not specific for Pythium but more general for application of the RNAi strategy in any organism

Line 324: delete ‘sequence’

Line 332: … can bind directly to mRNA…

Line 354: replace ‘,and’ by ‘;’

Line 368: is there a code for the WT strain? It makes sense to give it a code as this particular strain is proven to be amenable for CRISPR-Cas9 genome editing. And also as the source of the RNA seq data

PLOS authors have the option to publish the peer review history of their article (what does this mean?). If published, this will include your full peer review and any attached files.

Reviewer #1: No

Reviewer #2: No

Reviewer #3: No

Figure Files:

Data Requirements:

Reproducibility:

References:

---

## [Editor Report · Decision Letter 1]

29 Sep 2021

Dear Dr. Ye,

Thank you very much for submitting your manuscript "Specific interaction of an RNA-binding protein with the 3′-UTR of its target mRNA is critical to oomycete sexual reproduction" for consideration at PLOS Pathogens. As with all papers reviewed by the journal, your manuscript was reviewed by members of the editorial board and by several independent reviewers. The reviewers appreciated the attention to an important topic. Based on the reviews, we are likely to accept this manuscript for publication, providing that you modify the manuscript according to the recommendations shown below:

The authors have generally provided satisfactory changes to the manuscript that address the reviewer's main concerns. A few small improvements are recommended, however:

Line 87: "not effectively applied" seems the wrong thing to say. The implication is that the methods are not effective, but I see no evidence of this. The only thing that can be said that for sure is that few people have reported using those methods. This should be reworded. Otherwise, it looks like the authors are insulting the prior work.

Line 81: the meaning of "related genes" is unclear

Line 89: Cas9 has not been used successfully in P. infestans, only Cas12a has been used. You could reword this sentence to say CRISPR/Cas instead of CRISPR/Cas9.

Line 231: "100 nt after stop codon": surely these were not always exactly at this site. Do you mean within 100 nt of the stop codon? I would reword this, and perhaps indicate how far the site was on average from the polyA tail.

Table S2: The headings could be improved to indicate that 24, 48, 96 hr etc. are wild-type

Line 234: The wording "over 12-fold more" was confusing. This is just a language issue; there were 12-fold more genes differentially expressed at the 96 hr compared to the 24 hr timepoint. Also, why not just give (or also give) the number of genes?

Finally, reviewer 3 suggested that in Fig S6 that the GFP tag interferes with the function of PYU1_T003505 in the transformant and that therefore overexpression does not show a change in phenotype? In rebuttal, the authors give a reference showing that GFP did not interfere with argonaute function in another paper. However, the effect of the tag can be protein-specific; sometimes it has no effect, sometimes it does. I do not expect the authors to perform the type of experiment typically used to address this (expressing GFP from both the C and N-termini). However, they should acknowledge that their conclusion assumes that GFP is not affecting the function of the protein.

Sincerely,

Howard S. Judelson

Guest Editor

PLOS Pathogens

Bart Thomma

Section Editor

PLOS Pathogens

Kasturi Haldar

Editor-in-Chief

PLOS Pathogens

orcid.org/0000-0001-5065-158X

Michael Malim

Editor-in-Chief

PLOS Pathogens

orcid.org/0000-0002-7699-2064

The authors have generally provided satisfactory changes to the manuscript that address the reviewer's main concerns. A few small improvements are recommended, however:

Line 87: "not effectively applied" seems the wrong thing to say. The implication is that the methods are not effective, but I see no evidence of this. The only thing that can be said that for sure is that few people have reported using those methods. This should be reworded. Otherwise, it looks like the authors are insulting the prior work.

Line 81: the meaning of "related genes" is unclear

Line 89: Cas9 has not been used successfully in P. infestans, only Cas12a has been used. You could reword this sentence to say CRISPR/Cas instead of CRISPR/Cas9.

Line 231: "100 nt after stop codon": surely these were not always exactly at this site. Do you mean within 100 nt of the stop codon? I would reword this, and perhaps indicate how far the site was on average from the polyA tail.

Table S2: The headings could be improved to indicate that 24, 48, 96 hr etc. are wild-type

Line 234: The wording "over 12-fold more" was confusing. This is just a language issue; there were 12-fold more genes differentially expressed at the 96 hr compared to the 24 hr timepoint. Also, why not just give (or also give) the number of genes?

Finally, reviewer 3 suggested that in Fig S6 that the GFP tag interferes with the function of PYU1_T003505 in the transformant and that therefore overexpression does not show a change in phenotype? In rebuttal, the authors give a reference showing that GFP did not interfere with argonaute function in another paper. However, the effect of the tag can be protein-specific; sometimes it has no effect, sometimes it does. I do not expect the authors to perform the type of experiment typically used to address this (expressing GFP from both the C and N-termini). However, they should acknowledge that their conclusion assumes that GFP is not affecting the function of the protein.

Figure Files:

Data Requirements:

Reproducibility:

References:

---

## [Editor Report · Decision Letter 2]

3 Oct 2021

Dear Dr. Ye,

We are pleased to inform you that your manuscript 'Specific interaction of an RNA-binding protein with the 3′-UTR of its target mRNA is critical to oomycete sexual reproduction' has been provisionally accepted for publication in PLOS Pathogens.

Best regards,

Howard S. Judelson

Guest Editor

PLOS Pathogens

Bart Thomma

Section Editor

PLOS Pathogens

Kasturi Haldar

Editor-in-Chief

PLOS Pathogens

orcid.org/0000-0001-5065-158X

Michael Malim

Editor-in-Chief

PLOS Pathogens

orcid.org/0000-0002-7699-2064
---

## [Editor Report · Acceptance letter]

11 Oct 2021

Dear Dr. Ye,

We are delighted to inform you that your manuscript, "Specific interaction of an RNA-binding protein with the 3′-UTR of its target mRNA is critical to oomycete sexual reproduction," has been formally accepted for publication in PLOS Pathogens.

Best regards,

Kasturi Haldar

Editor-in-Chief

PLOS Pathogens

orcid.org/0000-0001-5065-158X

Michael Malim

Editor-in-Chief

PLOS Pathogens

orcid.org/0000-0002-7699-2064